# Targeted Unlearning with Single Layer Unlearning Gradient

Zikui Cai [1 2 *]   Yaoteng Tan [1 *]   M. Salman Asif [1]

## Abstract

Machine unlearning methods aim to remove sensitive or unwanted content from trained models, but typically demand extensive model updates at significant computational cost while potentially degrading model performance on both related and unrelated tasks. We propose Single Layer Unlearning Gradient (SLUG) as an efficient method to unlearn targeted information by updating a single critical layer using a one-time gradient computation. SLUG uses layer importance and gradient alignment metrics to identify the optimal layer for targeted information removal while preserving the model utility. We demonstrate the effectiveness of SLUG for CLIP, Stable Diffusion, and vision-language models (VLMs) in removing concrete (e.g., identities and objects) and abstract concepts (e.g., artistic styles). On the UnlearnCanvas benchmark, SLUG achieves comparable unlearning performance to existing methods while requiring significantly less computational resources. Our proposed approach offers a practical solution for targeted unlearning that is computationally efficient and precise. Our code is available at https://github.com/CSIPlab/SLUG.

## 1 Introduction

Modern large foundation models, including large language models (LLMs) (Leiter et al., 2024), text-to-image diffusion (Salimans & Ho, 2022; Yang et al., 2023) , and vision-language models (VLMs) (Zhang et al., 2024b; Liu et al., 2024b) leverage vast amounts of data for training. While these large scale datasets enhance performance, they also raise serious data privacy and legal compliance (gdp, 2016; Thiel, 2023) concerns as unwanted data can influence the trained models, resulting in harmful content genera-

tion (Thiel, 2023) and demands for unlearning. Completely abandoning trained large models and retraining them from scratch using scrutinized dataset is prohibitively expensive and wasteful. **Machine unlearning** (Cao & Yang, 2015; Nguyen et al., 2022; Chakraborty et al., 2024) is an attractive alternative, which refers to techniques designed to remove targeted information from a trained model.

A good machine unlearning technique should achieve **three main objectives: (1) Computational efficiency**. The naïve approach of re-training models achieves exact unlearning but at an enormous computational cost. An effective solution must minimize the computational overhead. **(2) Effective and robust unlearning**. Recent studies have raised concerns about the robustness of unlearning methods, showing that unlearned concepts can often be recovered through careful probing or adversarial techniques (Zhang et al., 2025; Petsiuk & Saenko, 2025; Che et al., 2025). The unlearning process must be robust against such recovery attempts. **(3) Targeted removal with minimal side effects**. The interconnected nature of learned representations means that removing one concept can lead to degraded performance on (un)related concepts and overall model performance (Amara et al., 2025). Unlearning should precisely target specific information while preserving general model capabilities.

Current unlearning methods often fail to meet all three objectives simultaneously. Traditional approaches like fine-tuning (FT) (Warnecke et al., 2023) and gradient ascent (GA) (Thudi et al., 2022) struggle to balance between effective forgetting and utility preservation. More recent techniques such as saliency unlearning (SalUn) (Fan et al., 2024) and selective synaptic dampening (SSD) (Foster et al., 2024) attempt to address this by identifying and updating only salient parameters. While these methods represent the state-of-the-art, they still face **key challenges: (1) High computational cost,** as they typically involve iterative gradient computation and parameter updates across the entire model. **(2) Limited generalizability and flexibility,** as they are often designed for single tasks (e.g. classification, image generation). **(3) Side effects,** as parameter updates spread over the entire model can cause unintended changes to (un)related concepts and degrade overall model performance. **(4) Human engineering,** as they require extensive hyperparameter tuning for learning rate, number of iterations, and thresholds for masking in parameter updates. We

---
[*]Equal contribution  [1]University of California Riverside, Riverside, CA, USA  [2]University of Maryland, College Park, MD, USA. Correspondence to: M. Salman Asif <sasif@ucr.edu>.

*Proceedings of the 42nd International Conference on Machine Learning*, Vancouver, Canada. PMLR 267, 2025. Copyright 2025 by the author(s).

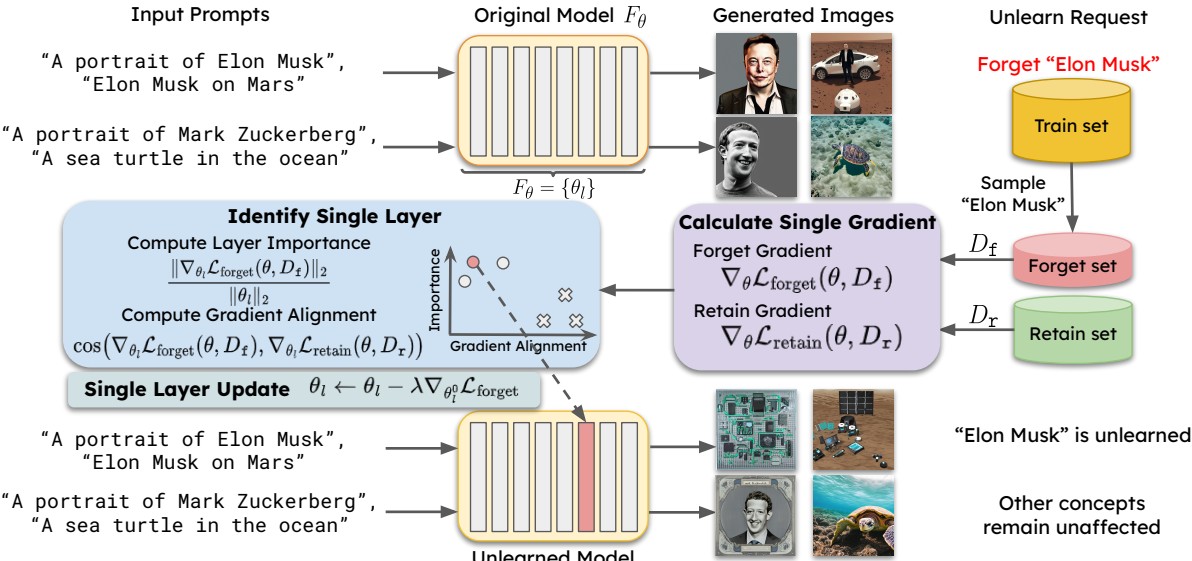

**Figure 1:** Overview of proposed **S**ingle **L**ayer **U**nlearning **G**radient (SLUG) framework. Given an unlearning request, we curate a forget set and retain set, then compute gradients w.r.t corresponding loss for one-time. We then identify the most critical layer to update that has larger importance for forgetting, and smaller gradient alignment between forget and retain. A binary search helps determine the step size $\lambda$ for model updating, ensuring targeted forgetting while retaining model utility.

present additional discussion on related work in Section A.

To address these challenges, we propose **Single Layer Unlearning Gradient (SLUG)** (see Figure 1 for illustration) to overcome the key challenges listed above. SLUG uses layer importance and gradient alignment metrics to identify a single optimal layer for targeted information removal while preserving the model utility. SLUG minimizes the computational cost by requiring a one-time gradient computation and single layer updates. Furthermore, SLUG concentrates changes in a single layer and introduces strong flexibility, making post-training large scale model updates more modular and efficient compared to existing methods.

We demonstrate the effectiveness of SLUG across CLIP, Stable Diffusion, and vision-language models in removing concrete (e.g., identities and objects) and abstract concepts (e.g., artistic styles). Our experiments demonstrate SLUG effectiveness across all three key objectives. For efficiency, we achieve state-of-the-art results on the UnlearnCanvas benchmark while requiring only a fraction of the computational resources and tiny storage. For precision, we show minimal impact on related concepts and image quality. In terms of robustness, we evaluate against recent vulnerabilities identified by Zhang et al. (2025) and Petsiuk & Saenko (2025), demonstrate the effectiveness of our method.

## 2 Single Layer Unlearning Gradient

The core principles of SLUG can be summarized as follows. (1) Compute one-time gradients for the forget and retain losses. (2) Identify a single layer with high importance to the forget set and low relevance to the retain set. (3) Update the targeted layer along a linear path using the computed forget gradient. Below we discuss details about the unlearning problem formulation, layer identification, unlearning via single gradient, and generalization to different models. We have included a pseudocode for SLUG in Section B.

### 2.1 Unlearning problem formulation

**Preliminaries.** Suppose we are given a model $F_\theta(D)$ with parameters $\theta$ trained on dataset $D$ with $N$ samples. Our goal is to remove the influence of a specific **forget set** $D_{\mathtt{f}} \subset D$, consisting of $N_{\mathtt{f}}$ samples, on $F_\theta(D)$. The challenge is to make this process more efficient than retraining the model on the **retain set** $D_{\mathtt{r}} = D \setminus D_{\mathtt{f}}$ with $N_{\mathtt{r}}$ samples. We seek to develop an unlearning algorithm $U$ that produces an unlearned model $F_{\theta_{\mathtt{f}}} = U(F_\theta(D), D, D_{\mathtt{f}})$, which is functionally equivalent to a model $F_{\theta_{\mathtt{r}}}(D_{\mathtt{r}})$ that is retrained only on $D_{\mathtt{r}}$. We can formulate the unlearning problem as minimizing some loss functions defined on the retain and forget sets:

$$\min_\theta \ \mathcal{L}_{\text{retain}}(\theta, D_{\mathtt{r}}) - \alpha\mathcal{L}_{\text{forget}}(\theta, D_{\mathtt{f}}), \qquad (1)$$

where $\alpha$ denotes a regularization parameters.

**Loss functions for vision-language alignment.** For traditional image classification models, cross-entropy loss can be directly used as both retain loss and forget loss. In this paper, we focus on large multi-modal foundation models such as CLIP (Radford et al., 2021), Stable Diffusion (Rombach et al., 2022), and vision-language models (Liu et al., 2024b) that rely on vision-language alignment. CLIP, in particu-

lar, is pivotal in advancing multi-modal models by aligning visual and textual representations through contrastive loss (Chopra et al., 2005). In unlearning, one of our goals is to break these learned alignments, so that one modality is non-retrievable with the corresponding modality.

For the retain set, we use the original contrastive loss that can be defined as

$$\mathcal{L}_{\text{retain}}(\theta, D_{\text{r}}) = \frac{1}{2N_{\text{r}}} \sum_{i=1}^{N_{\text{r}}} \left( \ell_{i2t}(i) + \ell_{t2i}(i) \right), \quad (2)$$

where

$$\begin{aligned} \ell_{i2t}(i) &= -\log \frac{\exp(\cos(\mathbf{v}_i, \mathbf{t}_i)/\tau)}{\sum_{j=1}^{N} \exp(\cos(\mathbf{v}_i, \mathbf{t}_j)/\tau)}, \\ \ell_{t2i}(i) &= -\log \frac{\exp(\cos(\mathbf{t}_i, \mathbf{v}_i)/\tau)}{\sum_{j=1}^{N} \exp(\cos(\mathbf{t}_i, \mathbf{v}_j)/\tau)}. \end{aligned} \quad (3)$$

Here, $\mathbf{v}_i$ is the normalized image embedding from the vision model $f_{\text{v}}$, and $\mathbf{t}_i$ is the normalized text embedding from the text model $f_{\text{t}}$. The temperature $\tau$ controls the sharpness of the softmax probability distribution, while cosine similarity is defined as $\cos(\mathbf{v}_i, \mathbf{t}_j) = \mathbf{v}_i \cdot \mathbf{t}_j$. Minimizing this contrastive loss aligns the vision and language representations in the embedding space.

For the forget set, we use the cosine embedding loss:

$$\mathcal{L}_{\text{forget}}(\theta, D_{\text{f}}) = \frac{1}{N_{\text{f}}} \sum_{i=1}^{N_{\text{f}}} 1 - \cos(\mathbf{v}_i, \mathbf{t}_j). \quad (4)$$

This loss directly pushes the embeddings of positive pairs away while not tampering with the embeddings of negative pairs. Using the original contrastive loss as forget loss will result in ineffective unlearning.

## 2.2 Single layer identification

Inspired by the findings that reveal distinct layers learned distinct features in deep networks from Zeiler & Fergus (2014); Olah et al. (2017); Ghiasi et al. (2022). We aim to identify and modify only the critical layers that contain features related to the unlearning tasks while avoiding changes to other layers, which capture abstract features unrelated to unlearning but essential for model utility. To minimize the impact on utility when modifying the model, we propose updating parameters along the unlearning direction within the "null space" of the retaining features. In other words, we focus on layers that minimally change retain set outputs while maximally changing the forget set outputs. This approach balances the impact on retained data while precisely targeting features for unlearning, enhancing the precision of model modification.

We quantify the importance of a layer $l$ to the forget set $D_{\text{f}}$ using the ratio of the $\ell_2$ norm of the forget loss gradients to the $\ell_2$ norm of the layer $l$ parameters $\theta_l$:

$$\text{Importance}(l) = \frac{\|\nabla_{\theta_l} \mathcal{L}_{\text{forget}}(\theta, D_{\text{f}})\|_2}{\|\theta_l\|_2}. \quad (5)$$

This choice of importance is inspired by the use of Fisher information matrix to measure the influence of model parameters (Foster et al., 2024; Kay, 1993; Hassibi et al., 1993; Kirkpatrick et al., 2017). Note that the Fisher information matrix for $\theta_l$ can be defined as

$$\mathcal{I}_D(\theta_l) = \mathbb{E}\left[ \left( \frac{\partial}{\partial \theta_l} \mathcal{L}(\theta; D) \right) \left( \frac{\partial}{\partial \theta_l} \mathcal{L}(\theta; D) \right)^{\mathsf{T}} \right], \quad (6)$$

where $\mathcal{L}$ denotes the log-likelihood loss (i.e., score) function (which can be the forget loss in our case). The diagonal elements reflect the sensitivity of the log-likelihood to the parameter changes, and the $\ell_2$ norm of the diagonal entries is identical to the $\ell_2$ norm of the forget loss gradient.

Importance of layer alone is insufficient for balancing unlearning and utility retention. We also ensure that forget gradients are nearly orthogonal to the retain gradients by minimizing the gradient alignment:

$$\text{Alignment}(l) = \cos\big( \nabla_{\theta_l} \mathcal{L}_{\text{forget}}(\theta, D_{\text{f}}), \nabla_{\theta_l} \mathcal{L}_{\text{retain}}(\theta, D_{\text{r}}) \big). \quad (7)$$

Small alignment between unlearn and retain gradients would prevent unlearning updates from negatively affecting the retain set.

To balance both objectives, we search for a Pareto-optimal set across all layers (Marler & Arora, 2010), maximizing importance while minimizing alignment of forget and retain gradients. Figure 2 illustrates the Pareto front for unlearning a target identity from CLIP ViT-B/32, where colored dots represent layers that achieve optimal trade-offs between these objectives—improving one metric necessarily worsens the other. We aim to find the optimal balance between unlearning and retention among these selected layers.

## 2.3 Unlearning in a single gradient direction

Existing unlearning methods calculate gradients at each iteration to update model parameters, which significantly increases computational complexity. Inspired by task arithmetic (Ilharco et al., 2023) and the linear nature of many optimization problems (LeCun et al., 2015), we observe that repeated gradient calculations can be redundant. Instead, we propose calculating the gradient only once for the initial model and updating the parameters $\theta_l$ of any layer $l$ in a weight-arithmetic fashion. Specifically, the weights are updated along a fixed gradient direction:

$$\theta_l^* \leftarrow \theta_l^{(0)} - \lambda^* \nabla_{\theta_l} \mathcal{L}_{\text{forget}}(\theta, D_{\text{f}}) \Big|_{\theta = \theta^{(0)}}, \quad (8)$$

where $\theta_l^*$ represents the parameters of layer $l$ for the unlearned model and $\theta_l^{(0)}$ represents the initial parameters. The

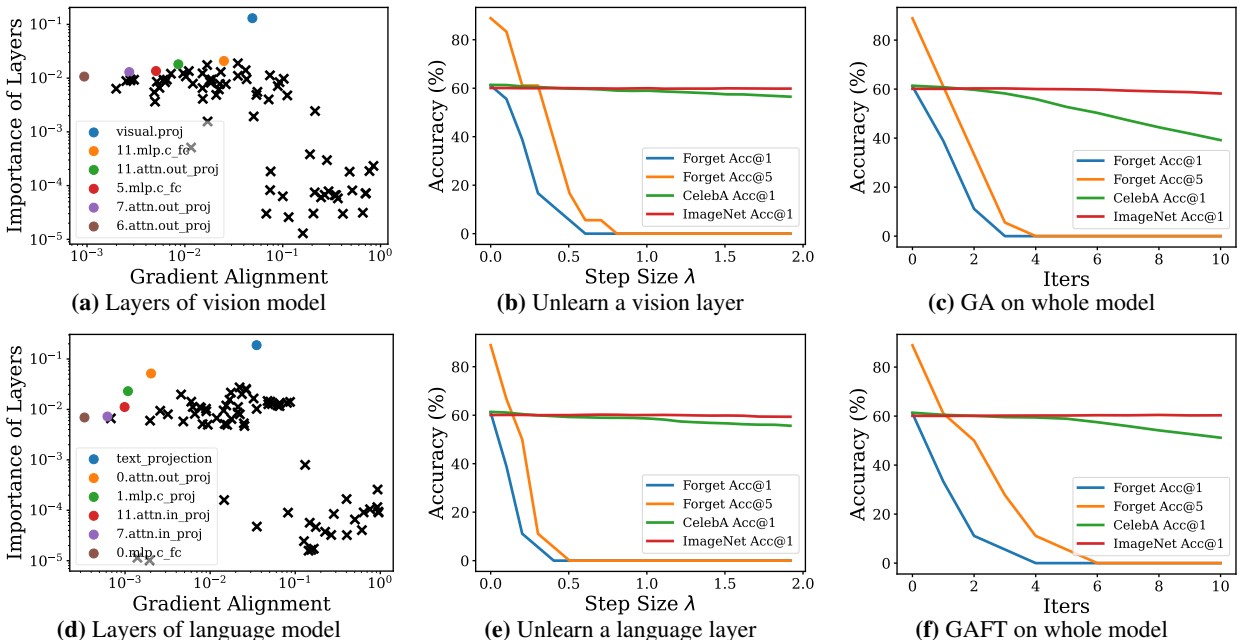

**Figure 2:** Layer identification (a,d) and unlearning with a single gradient (b,e). The first column shows gradient alignment and importance metrics for vision and language models from CLIP ViT-B-32, highlighting layers on the Pareto front for unlearning an identity. The second column demonstrates effective unlearning by updating identified layers along a single gradient direction without significantly impacting retain set performance. The third column shows that iterative methods (GA and GAFT) offer no advantage over a single gradient and require early stopping to prevent over-unlearning.

gradient $\nabla_{\theta_l} \mathcal{L}_{\text{forget}}(\theta, D_f)\big|_{\theta=\theta^{(0)}}$ is calculated only once, based on the forget loss $\mathcal{L}_{\text{forget}}$ evaluated on the forget set $D_f$. The step size $\lambda^*$ controls the update magnitude.

Updating weights of a layer along a fixed gradient direction is equivalent to linearizing the unlearning trajectory. This approach reduces computational complexity while ensuring effective unlearning. To select the appropriate step size $\lambda^*$, we perform a binary search along the linearized path, halting when the evaluation metric indicates satisfactory unlearning without harming performance on the retain set. For example, we stop at $\lambda \approx 0.75$ in Figure 2b, where the forget accuracy is near zero and test accuracy is high. This method strikes a balance between computational efficiency and precision, maintaining model utility while achieving unlearning goals.

### 2.4 Generalization to Stable Diffusion and VLMs

Following the effective unlearning in CLIP models, our technique can be further extended to foundation models built on CLIP encoders, such as Stable Diffusion (SD) and VLMs like LLaVA (Liu et al., 2024b;a).

**Unlearn Stable Diffusion.** Text-to-image (stable diffusion) models use a pretrained text encoder $f_t(\cdot)$ to project text prompts into high-dimensional vectors, which serve as guidance in the denoising process. A text-guided denoising step at time $t$ is written as

$$\mathbf{x}_{t-1} = \sqrt{\alpha_t}\left(\mathbf{x}_t - \gamma_t \nabla_{\mathbf{x}} \log p(\mathbf{x}_t|\mathbf{e})\right) + \sqrt{1-\alpha_t}\mathbf{z}_t, \quad (9)$$

where $\mathbf{x}_t$ is the noisy image, $\mathbf{z}_t$ is the random noise, $\alpha_t$

is a time-dependent noise balance factor, $\gamma_t$ is the guidance scale, $\mathbf{e} = f_t(\texttt{txt})$ is the text embedding, and $\nabla_{\mathbf{x}} \log p(\mathbf{x}_t|\mathbf{e})$ is the gradient of the log-probability of the noisy image conditioning over the text embedding (also known as the conditional score function), guiding the denoising process. We achieve single-layer unlearning on SD by applying SLUG to the text encoder (i.e., CLIP), enabling a layer-level flexible plug-in unlearner inspired by Zhang et al. (2024c).

**Unlearn Vision-Language Models.** VLMs enable LLMs to process visual modality by employing pretrained vision encoder $f_{\mathbf{v}}(\cdot)$. LLaVA-1.5 uses a pretrained CLIP vision encoder to extract the visual feature from images $\mathbf{e} = f_{\mathbf{v}}(\texttt{img})$, which are then projected to visual tokens $\mathbf{H_v} = \mathbf{W} \cdot \mathbf{e}$ through an MLP $\mathbf{W}$. These tokens are then concatenated with language tokens $\mathbf{H_q}$ as input $\mathbf{H} = [\mathbf{H_v}; \mathbf{H_q}]$ to the language model. Since VLMs rely solely on the vision encoder to understand images, similarly to unlearn SDs, we apply SLUG on the vision encoder of VLMs to influence the downstream text generation.

## 3 Experiments and Results

In this section, we first provide a brief overview of the experimental setup for each experiment. We then present key results demonstrating the effectiveness of SLUG in unlearning CLIP, Stable Diffusion, and vision-language models.

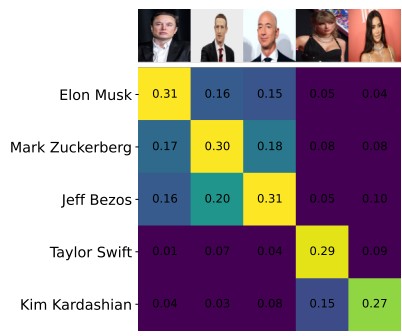



**(a)** Original cosine similarity matrix

**(b)** Cosine similarity matrix after unlearning

**Figure 3:** Cosine similarity matrix of image-text pairs before & after unlearning "Elon Musk" as an example. (a) original CLIP correctly associate images and text of distinct identities with high similarity. (b) after unlearning, the image-text pair of "Elon Musk" is no longer matched, while other identities are only slightly affected.

**Table 1:** Performance comparison of different unlearning methods on CLIP zero-shot classification. FA@{1, 5} stands for top-{1, 5} forget accuracy (%), i.e., accuracy of unlearned identity. TA_IN@1 and TA_CA@1 stands for the top-1 test accuracy (%) on ImageNet and CelebA dataset, respectively. $K$ and $k$ denotes the number of epochs for training and iterations for unlearning, respectively ($K = 32$ and $k = 10$ in our experiments). $N$ is the training set size, which is much larger than our sampled forget set ($N_f$) and retain set ($N_r$).

| METHOD | FA@1 ($\downarrow$) | FA@5 ($\downarrow$) | TA_IN@1 ($\uparrow$) | TA_CA@1 ($\uparrow$) | COMPUTE TIME ($\mathcal{O}$) |
|---|---|---|---|---|---|
| ORIGINAL | 73.05 | 92.22 | 60.12 | 61.38 | $\mathcal{O}(K \cdot N)$ |
| | | LEARNING RATE | $= 10^{-6} / 10^{-7}$ | | |
| FT (WARNECKE ET AL., 2023) | 66.08/70.50 | 90.10/92.22 | 60.36/60.26 | 60.70/61.35 | $\mathcal{O}(k \cdot N_r)$ |
| GA (THUDI ET AL., 2022) | 0.00/0.00 | 0.00/4.91 | 35.88/60.03 | 24.92/53.86 | $\mathcal{O}(k \cdot N_f)$ |
| GAFT ((1)) | 0.00/2.67 | 0.00/15.89 | 55.52/60.13 | 25.71/55.22 | $\mathcal{O}(k \cdot (N_f + N_r))$ |
| SALUN (FAN ET AL., 2024) | 0.00/3.33 | 0.00/15.69 | 55.45/60.26 | 26.11/55.81 | $\mathcal{O}(N_f) + \mathcal{O}(k \cdot (N_f + N_r))$ |
| SSD (FOSTER ET AL., 2024) | 0.00 | 0.00 | 51.84 | 35.96 | $\mathcal{O}(N_f + N_r)$ |
| SLUG (OURS) | 0.00 | 0.00 | 59.96 | 58.32 | $\mathcal{O}(N_f + N_r)$ |

## 3.1 Experiment setup

**Unlearning scenarios.** Considering practicality, we explore three key unlearning scenarios: (1) unlearning celebrity identity information to address privacy concerns, (2) unlearning copyrighted content to comply with legal standards, and (3) unlearning artistic styles and object concepts in UnlearnCanvas (Zhang et al., 2024d). Our focus is on large-scale foundation models, including CLIP (Radford et al., 2021), Stable Diffusion (Rombach et al., 2022), and VLM (Liu et al., 2024b).

**Models.** We performed experiments on CLIP, Stable Diffusion (SD), and VLM to demonstrate the broad applicability of our method. For CLIP, we used architectures ranging from ViT-B-32 to EVA01-g-14, trained on LAION-400M dataset (Schuhmann et al., 2021), and model weights sourced from the OpenCLIP repository (Cherti et al., 2023). For SD, we used SDv1.5 and SDv2.1 from StabilityAI, which employs CLIP-ViT-H-14 as text encoder, trained on the LAION-5B dataset. For VLM, we used the improved LLaVA-v1.5-7B model from HuggingFace, which employs a CLIP ViT-L/14-336px as vision encoder, from OpenAI. We provide a summary of model sizes in Appendix F.

**Datasets.** We used publicly-available datasets to construct the forget, retain, and validation sets. For identity unlearning, we curated the forget set by filtering the LAION-400M dataset to isolate 1,000 to 6,000 image-text pairs per identity. The retain set consists of a single shard from LAION-400M, containing approximately 7,900 images (due to expiring URLs). To assess unlearning effectiveness, we used the CelebA dataset (Liu et al., 2015), sampling 100 frequently appearing celebrities from LAION-400M. The utility of post-unlearning models were evaluated with ImageNet dataset. *UnlearnCanvas* was used to test unlearning of artistic styles and objects in Stable Diffusion.

**Evaluation metrics.** For CLIP, we measure unlearning performance using Forget Accuracy, defined as the zero-shot classification accuracy on unlearned content. Following the standard zero-shot paradigm (Radford et al., 2021), predictions are based on the highest cosine similarity between image and text embeddings. Utility is assessed via zero-shot accuracy on ImageNet and CelebA. For SD, we employ established metrics from *UnlearnCanvas*. For VLM, we define Forget Accuracy as ratio: number of currently predicted instances / total number of instance, detailed in Section 3.4.

**Hyperparameters.** Our SLUG framework requires no manual hyperparameter tuning. We use binary search to determine the step size $\lambda$ for the one-step unlearning update (see Algorithm 3) that optimizes the trade-off between unlearning and retention metrics on a small validation subset. Across all experiments, we fix the number of binary search steps to $S = 10$. For validation at each search step, we use 5% of the test set for CLIP, 10 test-time generated images (not present in the forget training set) for SD, and a 10-image subset per identity for VLM unlearning. We further discuss the trade-off between validation size and unlearning performance in Appendix B.1.

**Comparing methods.** We compare with the state-of-the-art methods along with classical methods. For CLIP unlearning, we compare with classical fine tuning (FT) (Warnecke et al., 2023), gradient ascent (GA) / negative gradient (NG) (Thudi et al., 2022), and recent salient parameters-based SalUn (Fan et al., 2024), and SSD (Foster et al., 2024). We also compare with a two-stage GAFT approach (Fan et al., 2024), which first performs GA for $k$ steps on the forget set, then fine-tunes for $k$ steps on the retain set. For SD unlearning, we compare with 9 methods reported in *UnlearnCanvas*, detailed in Section 3.3.

### 3.2 Unlearning for CLIP

We demonstrate that modifying a single layer suffices to unlearn an identity or concept while preserving overall model utility. Figure 3 shows an example of unlearning Elon Musk from CLIP. Before unlearning (Figure 3a), image-text pairs of Elon Musk exhibit high cosine similarity, while after unlearning (Figure 3b), this similarity drops significantly, leaving other identities unaffected. Additional results for multiple identities and CLIP architectures in Section C (Appendix) further confirm the generalizability of our approach.

A key strength of SLUG is its ability to maintain performance on non-targeted tasks/data. Table 1 shows zero-shot classification accuracy on ImageNet and CelebA, where our method outperforms alternatives in both unlearning effectiveness and utility retention. Unlike iterative methods requiring extensive hyperparameter tuning, SLUG performs a single gradient computation ($\mathcal{O}(N_{\mathrm{f}} + N_{\mathrm{r}})$), avoiding the trade-offs seen in gradient-based approaches, where high learning rates compromise utility and low rates lead to ineffective unlearning.

**Localizing layers.** SLUG efficiently localizes critical layers for updates, reducing the search space from hundreds to just a few Pareto-optimal layers. Figure 2 highlights the critical layers for unlearning within a CLIP model, these layers balance layer importance (sensitivity to forget loss) and gradient alignment (minimizing impact on retained data). Colored dots represent Pareto-optimal layers, exhibiting high importance scores and low gradient alignment. By

performing a binary search for a step size that minimizes forget accuracy, SLUG effectively preserves model utility, as demonstrated in Figures 2b and 2e.

### 3.3 Unlearning for Stable Diffusion

In this section, we first present a qualitative evaluation on identity unlearning in SD, later, we present a comprehensive quantitative evaluation of SLUG on the established benchmark *UnlearnCanvas*.

**Unlearning identity.** We demonstrate the scenario of removing personal information on the latest SDv2.1. SLUG ensures SD from generating content related to the erased identity when given prompts corresponding to that identity. Figure 4 presents examples of images generated by SDs before and after unlearning with different methods. Our method interestingly maps the targeted identity "Elon Musk" to electronic circuits, consistently across various prompts, without compromising the model image generation on non-targeted concepts (suffering from ripple effects (Amara et al., 2025)). In contrast, other methods not only struggle with generating images of other identities (e.g., Mark Zuckerberg) but also degrade the quality of generated images on non-targeted concepts. In Section D, we provide additional results on unlearning more celebrity IDs, and other scenarios, including copyright-protected and novel concepts erasure.

**Evaluation on UnlearnCanvas benchmark.** To further demonstrate the unlearning effectiveness and efficiency of SLUG, we also evaluate its performance on the latest bench mark *UnlearnCanvas* (Zhang et al., 2024d), which focused on unlearning artistic style and object concepts in SDs. It introduces a comprehensive set of metrics, including **UA** (Unlearn Accuracy) for unlearning effectiveness, **IRA** (In-domain Retain Accuracy) and **CRA** (Cross-domain Retain Accuracy) for utility retention. The benchmark targets unlearning styles and objects on an SDv1.5 model fine-tuned to generate 20 different objects in 60 distinct styles, and focuses on unlearning one object/style at a time, yielding 80 unlearned models for evaluation. For dataset generation, the benchmark inputs the fine-tuned SD with the prompt: "A [object name] in [style name] style," to generate 20 images for each object-style pair, resulting in 24,000 images in total (as there are 1,200 object-style pairs). We curate the forget set for unlearning each style/object using the associated images from the *UnlearnCanvas* (i.e., 400 images per style and 1200 images per object).

In Table 2, we report the unlearning performance of SLUG on benchmark metrics, along with other state-of-the-art unlearning methods reported in *UnlearnCanvas*. Our method minimizes storage and computational time by only requiring the gradient values of a few layers on the Pareto front to be stored, and performing a one-step update along the gradi-

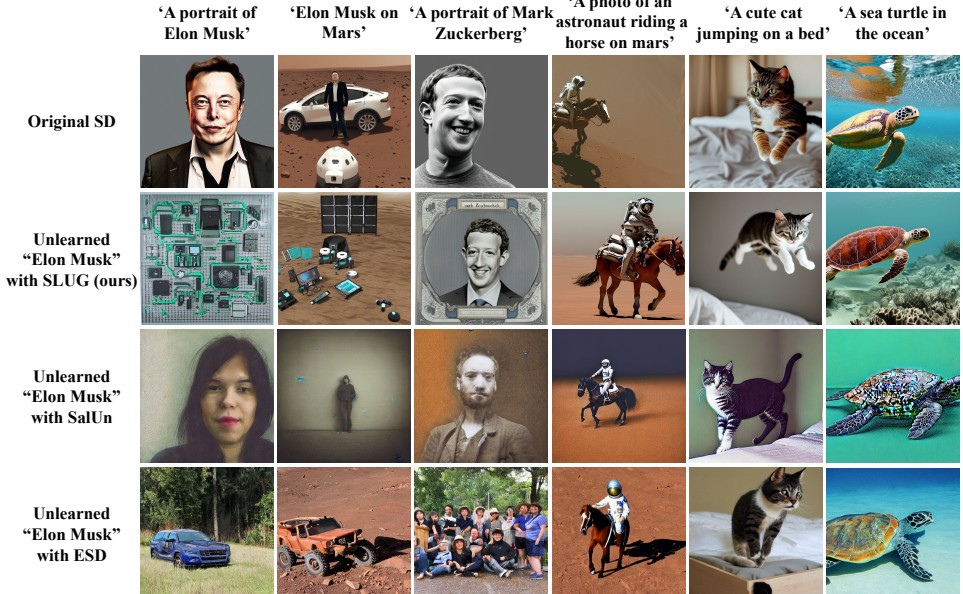

**Figure 4:** Images generated by different SDs using column captions as prompts. First row: images generated by the original pretrained SD. Second row: outputs of the SD after unlearning "Elon Musk" using SLUG. Bottom two rows: outputs of the SDs after unlearning "Elon Musk" using SalUn and ESD. While all methods unlearned the targeted identity, SLUG is superior on preserving the utility of the original model.

**Table 2:** Performance overview of different unlearning methods on UnlearnCanvas. The best performance for each metric is highlighted in green, and significantly underperforming results, in benchmark criteria, are marked in red. Our method SLUG shows no significant underperforming, and achieves the best trade-off among unlearning, retaining, and efficiency.

| METHOD | EFFECTIVENESS | | | | | | | EFFICIENCY | | | GAP RATIO |
| | STYLE UNLEARNING (%) | | | OBJECT UNLEARNING (%) | | | FID (↓) | TIME | MEMORY | STORAGE | MEAN |
| | UA (↑) | IRA (↑) | CRA (↑) | UA (↑) | IRA (↑) | CRA (↑) | | (s) (↓) | (GB) (↓) | (GB) (↓) | (%) (↓) |
|---|---|---|---|---|---|---|---|---|---|---|---|
| ESD (GANDIKOTA ET AL., 2023) | 98.58 | 80.97 | 93.96 | 92.15 | 55.78 | 44.23 | 65.55 | 6163 | 17.8 | 4.3 | 18.17 |
| FMN (ZHANG ET AL., 2024A) | 88.48 | 56.77 | 46.6 | 45.64 | 90.63 | 73.46 | 131.37 | 350 | 17.9 | 4.2 | 1.81 |
| UCE (GANDIKOTA ET AL., 2024) | 98.4 | 60.22 | 47.71 | 94.31 | 39.35 | 34.67 | 182.01 | 434 | 5.1 | 1.7 | 1.72 |
| CA (KUMARI ET AL., 2023) | 60.82 | 96.01 | 92.7 | 46.67 | 90.11 | 81.97 | 54.21 | 734 | 10.1 | 4.2 | 2.44 |
| SALUN (FAN ET AL., 2024) | 86.26 | 90.39 | 95.08 | 86.91 | 96.35 | 99.59 | 61.05 | 667 | 30.8 | 4.0 | 2.78 |
| SEOT (LI ET AL., 2024B) | 56.90 | 94.68 | 84.31 | 23.25 | 95.57 | 82.71 | 62.38 | 95 | 7.34 | 0.0 | 0.46 |
| SPM (LYU ET AL., 2024) | 60.94 | 92.39 | 84.33 | 71.25 | 90.79 | 81.65 | 59.79 | 29700 | 6.90 | 0.0 | 84.73 |
| EDIFF (WU ET AL., 2024) | 92.42 | 73.91 | 98.93 | 86.67 | 94.03 | 48.48 | 81.42 | 1567 | 27.8 | 4.0 | 5.37 |
| SHS (WU & HARANDI, 2024) | 95.84 | 80.42 | 43.27 | 80.73 | 81.15 | 67.99 | 119.34 | 1223 | 31.2 | 4.0 | 4.62 |
| SLUG (OURS) | 86.29 | 84.59 | 88.43 | 75.43 | 77.5 | 81.18 | 75.97 | 39 | 3.61 | 0.04 | 0.15 |
| BEST (HYPOTHETICAL) | 98.58 | 96.01 | 98.93 | 94.31 | 96.35 | 99.59 | 54.21 | 39 | 3.61 | 0.00 | 0.00 |

ent for unlearning. Despite being extremely efficient, our method does not suffer from significant performance degradation in any metric or task in *UnlearnCanvas*, as there is no red mark for *SLUG* row in Table 2. Our method achieves excellent trade-off between unlearning and retaining accuracy. For qualitative evaluation, we provide visual examples of style and object unlearning in Appendix D.

*Unified metric for UnlearnCanvas benchmark.* To summarize the different performance metrics of each method with a single quantity, we first define the Gap Ratio (GR) for metric $s$ and method $m$ as

$$\text{GR}(s, m) = \frac{|s_m - s_{\text{best}}|}{s_{\text{best}}}, \quad (10)$$

where $s_m$ and $s_{\text{best}}$ represent the metric for method $m$ and the best performing method, respectively. Intuitively, GR

represents the normalized distance of a method from the best performing method; lower values indicate performance closer to the best (hypothetical reference) method (see the BEST row of Table 2). To provide the summary statistics for each method (row) in Table 2, we compute the GR for UA, IRA, CRA for both style and object unlearning, FID, Time, and the sum of memory and storage (which are strongly related resources in practical deployments). This provides us a 9-dimensional GR vector for each method. We then report the mean of GR vector in the GAP RATIO column in Table 2, which is proportional to the $\ell_1$ distance between the GR vector of each method and the BEST (HYPOTHETICAL) method. The results indicate that SLUG has the smallest gap from the best reference method, further demonstrating the superior trade-off between effectiveness and efficiency

**Table 3:** Quantitative evaluation of SLUG for identity unlearning in LLaVA-1.5-7B vision-language model. The table shows forget accuracy and performance retention on standard VLM benchmarks across 10 celebrity identities. SLUG achieves effective unlearning with average forget accuracy dropping from 99.50% to 2.8%, while maintaining competitive performance on utility benchmarks compared to the original model, demonstrating targeted concept removal with minimal impact on general model capabilities.

| UNLEARN IDENTITY | FORGET ACCURACY (↓) | VLM BENCHMARK SCORE (↑) | | | |
|---|---|---|---|---|---|
| | | MME COGNITION | MME PERCEPTION | GQA | MMBENCH (%) |
| - | 99.50 | 323.57 | 1481.21 | 61.28 | 62.97 |
| ELON MUSK | 3.0 | 298.57 | 1354.61 | 60.70 | 61.34 |
| TAYLOR SWIFT | 2.0 | 334.64 | 1336.09 | 60.72 | 60.14 |
| MARK ZUCKERBERG | 7.0 | 343.57 | 1209.55 | 58.21 | 56.01 |
| JEFF BEZOS | 3.0 | 314.64 | 1315.32 | 60.40 | 61.43 |
| KANYE WEST | 4.0 | 314.64 | 1365.53 | 61.17 | 61.68 |
| TOM CRUISE | 0.0 | 351.79 | 1413.04 | 61.13 | 61.86 |
| KIM KARDASHIAN | 6.0 | 286.43 | 1249.54 | 60.42 | 60.14 |
| BARACK OBAMA | 0.0 | 288.57 | 1269.45 | 60.68 | 61.08 |
| LADY GAGA | 3.0 | 270.36 | 1178.45 | 58.55 | 55.58 |
| BRUCE LEE | 0.0 | 323.57 | 1266.21 | 60.44 | 60.22 |
| AVERAGE | 2.8±2.44 | 312.68±26.57 | 1295.78±73.78 | 60.24±1.02 | 59.95±2.28 |

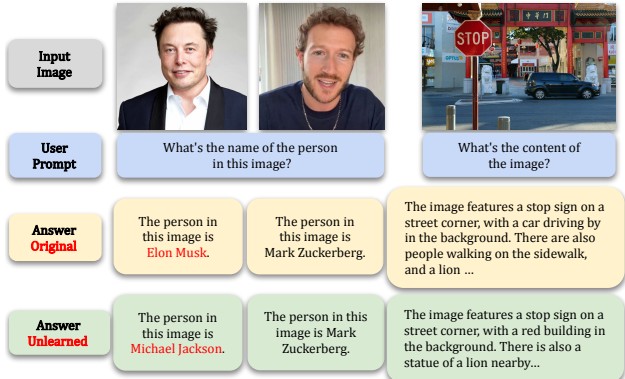

**Figure 5:** Unlearning "Elon Musk" on LLaVA-1.5 with SLUG. After unlearning, the model fails to identify "Elon Musk", whereas other identities/concepts remain unaffected.

that SLUG achieves. In Appendix D.4, we provide a detailed breakdown in terms of the effectiveness and efficiency aspects, along with GR evaluation using $\ell_1$ and $\ell_2$ norms.

**Robust evaluation.** Our results demonstrate that SLUG is robust to key unlearning vulnerabilities revealed in Petsiuk & Saenko (2025); Zhang et al. (2025), which include blackbox attacks that utilize the prompt arithmetic property of Stable Diffusion and model weight quantization. SLUG effectively resists concept arithmetic attacks, causes minimal ripple effects on related concepts (Amara et al., 2025), and maintains performance even under 8-bit weight quantization. The precise modification of a single layer allows for targeted concept removal, ensuring controlled downstream text-guided generation tasks while preserving model utility. Detailed experiment setup and results are provided in Section D.1. Additionally, we provide robustness evaluations for SLUG against whitebox prompt attacks (Zhang et al.,

2024e; Chin et al., 2024) in Section D.2.

### 3.4 Unlearning for VLMs

In this section, we present the results for SLUG on removing celebrity identity from VLMs. As there is a lack of an established VLM unlearning benchmark, we sample 10 different targeted identities from the CelebA, with 100 images per identity, resulting in 1000 images in total, to create a comprehensive validation set. We individually unlearn each identity from the original LLaVA-1.5-7B (Liu et al., 2024a), and evaluate the **Forget Accuracy** (FA) as

$$\text{FA} = \frac{\text{number of misidentified images}}{\text{total number of images}}. \quad (11)$$

For identification criteria, we input images associated with the targeted identity combined with the question prompt *"What is the name of the person in the image?"* to the model, and check whether the model answer matches the corresponding celebrity name. To evaluate the utility retention of unlearned models, we employ established VLM utility benchmarks: MME (Fu et al., 2023) GQA (Hudson & Manning, 2019), and MMBench (Liu et al., 2025b). These VLM benchmarks quantify performance of vision-language tasks, which cover a broad set of coarse-to-fine-grained questions on visual recognition and visual reasoning, characterizing the utility of a VLM. The results in Table 3 highlight that SLUG achieves effective unlearning while maintaining performance comparable to the original pretrained model, validating its effectiveness in the VLM context. Figure 5 provides a qualitative evaluation of our method. Our results demonstrate that SLUG successfully unlearned targeted identities from the VLM, while preserving utility.

**Table 4:** Ablation studies on parameter selection strategies for CLIP unlearning. Both layer importance and gradient alignment are essential for selecting layer to perform effective unlearning and utility retention under the one-step unlearning update framework.

| PARAMETER SELECTION | FA@1 ($\downarrow$) | FA@5 ($\downarrow$) | TA_IN@1 ($\uparrow$) | TA_CA@1 ($\uparrow$) |
|---|---|---|---|---|
| "SALUN" (DISTRIBUTED WEIGHTS, IMPORTANCE ONLY) | 4.44 | 11.33 | 48.23 | 37.38 |
| SINGLE LAYER IMPORTANCE ONLY | 0.0 | 0.0 | 21.04 | 42.00 |
| SINGLE LAYER ALIGNMENT ONLY | 0.0 | 5.56 | 31.08 | 54.16 |
| SINGLE LAYER AT RANDOM | 0.0 | 6.91 | 33.38 | 52.90 |
| ALL PARETO FRONT LAYERS | 0.0 | 0.0 | 59.92 | 51.64 |
| ALL LAYERS | 0.0 | 0.0 | 59.70 | 53.74 |
| SLUG (TABLE 1) | **0.0** | **0.0** | **59.96** | **58.32** |

## 4 Ablation studies

We conduct ablation studies to evaluate different parameter selection strategies and the effect of updating multiple layers under our one-step unlearning framework. Following the setup in Table 1, we target five identities for unlearning CLIP and assess forget accuracy (FA) for unlearning effectiveness, as well as zero-shot test accuracy on ImageNet (TA_IN) and CelebA (TA_CA) for utility retention assessment. The results are summarized in Table 4.

*SalUn* row shows that selecting weights across the entire network using only importance (SalUn-like strategy) performs worse than SLUG, with higher FA and lower TA. SINGLE LAYER IMPORTANCE row shows that selecting a single layer based on gradient importance alone reduces FA to 0, but significantly lowers TA_IN and TA_CA, revealing utility loss. SINGLE LAYER ALIGNMENT row shows that using alignment alone yields FA@5=5.56%, with notable utility degradation, indicating ineffective unlearning. SINGLE LAYER AT RANDOM row, where we randomly select a layer without guidance, performs the worst overall. These results highlight that both gradient importance and alignment are crucial for balancing unlearning effectiveness and utility retention, further justifying that SLUG achieves the best trade-off under the one-step unlearning framework. Beyond single layer, we also explore updating all Pareto-optimal layers and all model layers, as reported in ALL PARETO FRONT LAYERS and ALL LAYERS rows. While multi-layer updates improve FA by ~2–3%, they significantly increase computational cost. Furthermore, Figure 8 shows that selecting and updating a single layer per concept enables modular unlearning of multiple identities simultaneously, without requiring multi-layer updates for every concept.

## 5 Limitations

While SLUG represents the first endeavor to achieve unlearning through single-layer updates, it presents a trade-off between efficiency and effectiveness. SLUG demonstrates competitive performance with significant computational advantages, but does not achieve state-of-the-art results simultaneously across all metrics, tasks, and benchmarks. This paper does not provide a rigorous theoretical explanation for the success of single-layer updates for unlearning, and a formal understanding of when and why SLUG is effective remains an open question. This paper primarily focused on vision-language models; we did not explore the applicability of SLUG to large language models (LLMs), which operate purely in the language modality. While our experiments demonstrate that the models seem to forget/unlearn targeted identities and concepts with simple and efficient manipulation by SLUG; we cannot claim that those identities and concepts are completely erased/removed from the models by SLUG. Adversarial attacks and prompts can potentially retrieve the *unlearned* information. SLUG (like many other unlearning methods) lacks robustness against white-box adversarial attacks and its susceptibility to (whitebox and blackbox) relearning attacks needs further examination.

## 6 Conclusion

SLUG demonstrates that effective machine unlearning can be achieved through targeted single-layer modifications, offering a practical solution to the computational bottlenecks that have hindered large-scale model deployment and editing. Our results across CLIP, Stable Diffusion, and vision-language models reveal that the distributed nature of learned representations does not preclude precise, localized interventions—a finding that challenges conventional assumptions about the necessity of whole-model retraining or extensive parameter updates. The efficiency of SLUG opens new possibilities for dynamic model adaptation, where rapid response to removal requests is critical. The robustness of unlearning methods remains underexplored; in particular, their resilience against sophisticated prompting strategies or recovery attacks and their stability across different model architectures and deployment conditions. Future work should prioritize investigating these robustness challenges to establish unlearning as a reliable model editing technology.

### Acknowledgments

This work is supported in part by an NSF grant (CCF-2046293) and a UC SoCal HUB seed award.

## Impact Statement

This paper presents work whose goal is to advance the efficiency of machine unlearning for large-scale foundation models. By improving the ability to selectively remove data influence, our method contributes to trustworthy AI, addressing privacy concerns and regulatory compliance. While there are many potential societal consequences of our work, none that we feel must be specifically highlighted here.

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

## Appendix A   Related work

**Machine unlearning** (Cao & Yang, 2015; Nguyen et al., 2022) has recently emerged as a critical area of research, driven by privacy concerns and regulatory requirements (gdp, 2016). Existing approaches mainly focus on a single task, like image classification (Liu et al., 2024c; neu, 2023; Guo et al., 2020; Goel et al., 2022; Chien et al., 2022; Golatkar et al., 2020b;a; Chundawat et al., 2023; Kurmanji et al., 2023; Jia et al., 2023; Shaik et al., 2024; Fan et al., 2024; Foster et al., 2024), image generation (Li et al., 2024a; Gandikota et al., 2023; Zhang et al., 2024a; Gandikota et al., 2024; Kumari et al., 2023; Li et al., 2024b; Lyu et al., 2024; Wu et al., 2024; Wu & Harandi, 2024), and LLMs text generation (Yao et al., 2024; Liu et al., 2025a). In this work, we propose a generic approach that is applicable to a wide range of multi-modal models including CLIP (Radford et al., 2021) for zero-shot image classification, stable diffusion models (Rombach et al., 2022) for text-to-image generation, and vision-language models (Liu et al., 2024b) for visual question answering.

For text-to-image diffusion models, particularly Stable Diffusion (SD), the evolution of unlearning approaches reveals increasing sophistication. Early methods such as ESD (Gandikota et al., 2023) and CA (Kumari et al., 2023) focused on modifying the UNet architecture through fine-tuning with negative guidance, but these approaches often resulted in widespread parameter updates across multiple layers, potentially compromising generation fidelity. More recent work has explored more targeted and efficient interventions. UCE (Gandikota et al., 2024) introduced a training-free unified approach using closed-form solutions for simultaneous debiasing, style erasure, and content moderation. FMN (Zhang et al., 2024a) achieved rapid concept removal through attention re-steering loss, redirecting generation from unwanted concepts to pretrained alternatives. SPM (Lyu et al., 2024) proposed an adapter-based approach using "concept-SemiPermeable Membranes" that can be flexibly transferred across different models without re-tuning. Other approaches include EDiff (Wu et al., 2024), which formulates unlearning as a constrained optimization problem to preserve model utility, and SEOT (Li et al., 2024b), which focuses on content suppression through text embedding manipulation and inference-time optimization. Despite these advances, existing methods still face challenges in balancing computational efficiency, generalization ability, and preservation of model utility, which our work aims to address through a principled single-layer approach.

**Saliency-based methods.** Recent advances in machine unlearning have seen the emergence of saliency-based approaches, which aim to identify and modify only the most relevant parameters for concept removal. In image classification, methods like SSD (Foster et al., 2024) employ synaptic importance measures to selectively dampen connections, while SalUn (Fan et al., 2024) takes a simple and heuristic threshold-based approach. In text-to-image generation, SalUn (Fan et al., 2024) extend its framework by replacing cross-entropy loss in the unlearning objective to diffusion loss, requiring careful tuning of a gradient threshold for parameter selection. Diff-quickfix (Basu et al., 2024) utilizes causal inference with CLIPSscore (Hessel et al., 2021) as a metric to pinpoint concept-salient model parameters. MACE (Lu et al., 2024) proposes tuning the prompt-related projection matrices of the cross-attention blocks in the UNet architecture using LoRA modules (Hu et al., 2022). Similarly, CRE (Dong et al., 2024) identifies concept-specific causal denoising time steps in UNet layers and performs representation editing on selected layer outputs.

While these saliency-based methods represent the existing efforts in improving the efficiency of unlearning, their scope remains confined to specific tasks, such as image classification or text-to-image generation. Moreover, their parameter modifications often span multiple layers, which limits interpretability and flexibility in practical scenario. In contrast, our approach aims to extend efficient unlearning to foundation models that cover a diverse range of tasks (e.g., CLIP, Stable Diffusion, and vision-language models). By restricting model edits to a layer-specific scope, our framework introduces modularity to machine unlearning, abstracting the process into distinct layer updates along gradient vectors for tailored unlearning requests.

## Appendix B   Algorithm pseudocode

In this section, we present the pseudocode for our method, SLUG, in Algorithm 1, the search process for Pareto-optimal layers in Algorithm 2, and the binary search for the optimal unlearning step size in Algorithm 3. In Appendix B.1, we discuss the relation between validation size, binary search time cost and unlearning effectiveness.

Our implementation for the corresponding experimental models (i.e., CLIP, Stable Diffusion, and VLM) and benchmarks (i.e., UnlearnCanvas) has been made publicly available at the anonimized repository: https://github.com/CSIPlab/SLUG.

---

**Algorithm 1 SLUG**: Single Layer Unlearning Gradient

---

**Require:**

    Forget set $D_{\mathtt{f}}$ and retain set $D_{\mathtt{r}}$ ;

    Original model $F_\theta$ with model weights $\theta$;

    The set of all layers in the model, as $L$;

    Forget loss function $\mathcal{L}_{\text{forget}}$ and retain loss function $\mathcal{L}_{\text{retain}}$;

    Evaluation metrics forget accuracy FA and test accuracy TA.

**Ensure:** Unlearned model parameters $\theta_{\mathtt{f}}$

1: Calculate and store $\nabla_\theta \mathcal{L}_{\text{forget}}(\theta, D_{\mathtt{f}}), \nabla_\theta \mathcal{L}_{\text{retain}}(\theta, D_{\mathtt{r}})$      *▷ Single gradient calculation*

2: **for** each layer $l$ in $L$ **do**

3:      Importance$(l) = \|\nabla_{\theta_l}\mathcal{L}_{\text{forget}}(\theta, D_{\mathtt{f}})\|_2 / \|\theta_l\|_2$      *▷ Calculate layer importance*

4:      Alignment$(l) = \cos\big(\nabla_{\theta_l}\mathcal{L}_{\text{forget}}(\theta, D_{\mathtt{f}}), \nabla_{\theta_l}\mathcal{L}_{\text{retain}}(\theta, D_{\mathtt{r}})\big)$      *▷ Calculate layer alignment*

5: **end for**

6: $P = \textbf{ParetoOpt}(L, \text{Importance}, \text{Alignment})$      *▷ Pareto optimal algorithm 2*

7: $Q \leftarrow \emptyset$      *▷ Set of layers and their performances*

8: **for** each layer $l$ in $P$ **do**

9:      $\lambda_0 = \text{Importance}(l)/10$      *▷ Initialize step size*

10:      $(\lambda, \text{FA}, \text{TA}) = \textbf{BinarySearch}(\lambda_0, l)$      *▷ Binary search algorithm 3*

11:      $Q \leftarrow Q \cup \{(l, \lambda, \text{FA}, \text{TA})\}$

12: **end for**

13: $\text{FA}_{\min} = \min_{(l,\lambda,\text{FA},\text{TA})\in Q} \text{FA}$      *▷ Identify minimum FA*

14: $Q_{\min} = \{(l, \lambda, \text{FA}, \text{TA}) \in Q \,|\, \text{FA} = \text{FA}_{\min}\}$      *▷ Filter sets with minimum FA*

15: $(l^*, \lambda^*, \text{FA}^*, \text{TA}^*) = \arg\max_{(\lambda,\text{FA},\text{TA})\in Q_{\min}}(\text{TA})$      *▷ Select set with highest TA*

16: **return** $\theta_{\mathtt{f}} = \theta - \lambda^* \nabla_{\theta_{l*}}\mathcal{L}_{\text{forget}}(\theta, D_{\mathtt{f}})$

---

**Algorithm 2 Pareto Optimal:** $P = \textbf{ParetoOpt}(L, \text{Importance}, \text{Alignment})$

---

**Require:**

    The set of all layers in the model, as $L$;

    Layer importance and gradient alignment of all layers

**Ensure:** The set of Pareto optimal layers

1: Initialize $P \leftarrow \emptyset$      *▷ Set of layers on the Pareto front is empty*

2: **for** each layer $l$ in $L$ **do**

3:      ParetoDominant $\leftarrow$ **true**

4:      **for** each layer $l'$ in $L \setminus l$ **do**

5:          **if** (Importance$(l') >$ Importance$(l)$ **and** Alignment$(l') <$ Alignment$(l)$) **then**

6:              ParetoDominant $\leftarrow$ **false**

7:              **break**

8:          **end if**

9:      **end for**

10:      **if** ParetoDominant **then**

11:          $P \leftarrow P \cup \{l\}$      *▷ Identified a Pareto optimal layer*

12:      **end if**

13: **end for**

14: **return** $P$      *▷ Return the set of Pareto optimal layers*

---

### B.1 Relation of validation size, runtime, and unlearning effectiveness

The runtime of a single `eval()` function, in Algorithm 3, increases linearly with the validation set size. In Table 5, we provide the eval runtime and effectiveness of SLUG versus different validation set sizes, following the setup of Table 1 on CLIP unlearning. Note that our original choice of 5% validation size already provides a good test accuracy on ImageNet, close to that of the original model (which achieves 60.12%). While increasing the validation size slightly improves utility

retention after unlearning, it also increases evaluation time proportionally. Furthermore, a smaller validation size (1%) reduces the eval time to 3 seconds at the expense of slightly reduced TA.

**Table 5:** Forget accuracy, test accuracy on ImageNet, and runtime of unlearned CLIP models under various validation sizes.

| VALIDATION SIZE | NUMBER OF IMAGES | EVAL FUNCTION COST (S) | FA@1 ($\downarrow$) | TA_IN@1 ($\uparrow$) |
|---|---|---|---|---|
| 1% | 500 | 3.05 | 0.0 | 58.83 |
| 5 % (ORIGINAL) | 2500 | 6.62 | 0.0 | 59.96 |
| 20 % | 10000 | 24.83 | 0.0 | 59.94 |
| 50 % | 25000 | 59.69 | 0.0 | 59.98 |
| 100 % | 50000 | 119.04 | 0.0 | 60.04 |

---

**Algorithm 3** Binary Search for Optimal Step Size: $(\lambda^*, \text{FA}^*, \text{TA}^*) = \textbf{BinarySearch}(\lambda_0, l)$

---

**Require:**
    Initial step size $\lambda_0$;
    Maximum number of search steps $S$;
    Model parameters $\theta$;
    Forget gradient of layer l: $G_l = \nabla_\theta \mathcal{L}_{\text{forget}}(\theta, D_{\text{f}})$
**Ensure:** Optimal $\lambda^*$, forget accuracy FA, test accuracy TA
  1: $\lambda_{\text{low}} \leftarrow 0$
  2: $\lambda_{\text{high}} \leftarrow \infty$
  3: $\lambda \leftarrow \lambda_0$
  4: $s \leftarrow 0$
  5: Initialize $P \leftarrow \emptyset$          ▷ *Performance set*
  6: **while** $s < S$ **do**
  7:      $\text{FA}, \text{TA} = \texttt{eval}(\theta - \lambda G_l)$
  8:      $P \leftarrow P \cup \{(\lambda, \text{FA}, \text{TA})\}$          ▷ *Store results*
  9:      **if** $\text{FA} > 0$ **then**
10:          $\lambda_{\text{low}} \leftarrow \lambda$          ▷ *Should increase step size to unlearn*
11:      **else**
12:          $\lambda_{\text{high}} \leftarrow \lambda$          ▷ *Should reduce step size to avoid over-unlearning*
13:      **end if**
14:      **if** $\lambda_{\text{high}} == \infty$ **then**
15:          $\lambda \leftarrow 2\lambda$
16:      **else**
17:          $\lambda \leftarrow (\lambda_{\text{low}} + \lambda_{\text{high}})/2$
18:      **end if**
19:      $s \leftarrow s + 1$
20: **end while**
21: $\text{FA}_{\text{min}} = \min_{(\lambda, \text{FA}, \text{TA}) \in P} \text{FA}$          ▷ *Identify minimum FA*
22: $P_{\text{min}} = \{(\lambda, \text{FA}, \text{TA}) \in P \,|\, \text{FA} = \text{FA}_{\text{min}}\}$          ▷ *Filter sets with minimum FA*
23: $(\lambda^*, \text{FA}^*, \text{TA}^*) = \arg\max_{(\lambda, \text{FA}, \text{TA}) \in P_{\text{min}}} (\text{TA})$          ▷ *Select set with highest TA*
24: **return** $\lambda^*, \text{FA}^*, \text{TA}^*$          ▷ *Select the set with lowest FA which has the highest TA*

---

## Appendix C  More evaluations on unlearning CLIP

### C.1  More examples on unlearning identities

Building on the experiment with the target identity "Elon Musk" in Section 3.2, we provide a more comprehensive evaluation across a broader set of sampled identities. These names, selected from the CelebA dataset, represent a diverse range of ethnicities and genders. Our method effectively identifies the key layers associated with each identity, enabling efficient unlearning from the CLIP model. Figure 6 shows that our approach successfully removes the target identities, as evidenced by a significant decrease in image-text alignment (cosine similarity). We defer the corresponding pareto-front plots, which indicating the identified layer with SLUG in Section G.

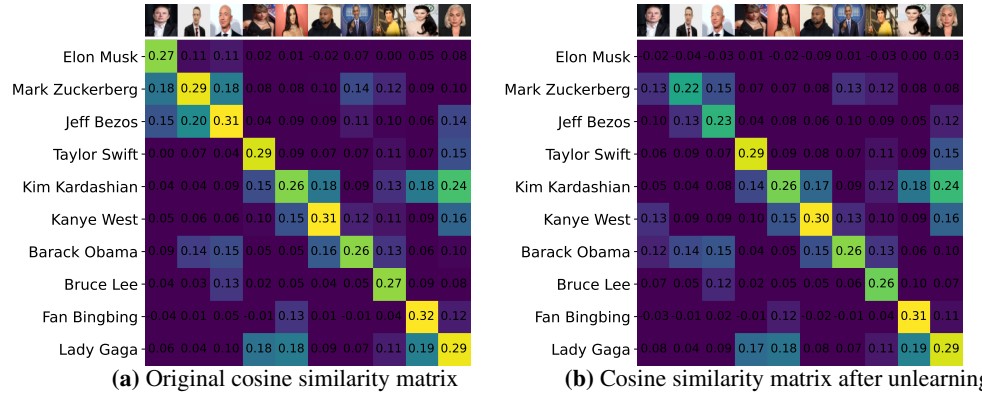

**(a)** Original cosine similarity matrix      **(b)** Cosine similarity matrix after unlearning

**Figure 6:** Cosine similarity matrix of image and text pairs before and after unlearning Elon Musk. After unlearning, the image and text pair of Elon Musk are not matched, while other persons are only slightly affected. Here the `vision attention out projection layer at the 9`$_{th}$` resblock` (associate with `9.attn.out_proj` in the pareto front legend) is unlearned. CLIP model: `ViT-B-16`

### C.2  Joint update for unlearning multiple identities

We extend SLUG to unlearn multiple identities simultaneously by computing gradients for each identity's forget set and identifying the most significant layers for joint updates. Following the update scheme in Section 2, we initialize step sizes separately for each identity and refine them via binary search based on unlearning effectiveness. Figure 7 demonstrates successful unlearning of (a) `Elon Musk, Mark Zuckerberg` and (b) `Elon Musk, Taylor Swift`, showcasing SLUG's ability to handle multiple unlearning tasks efficiently.

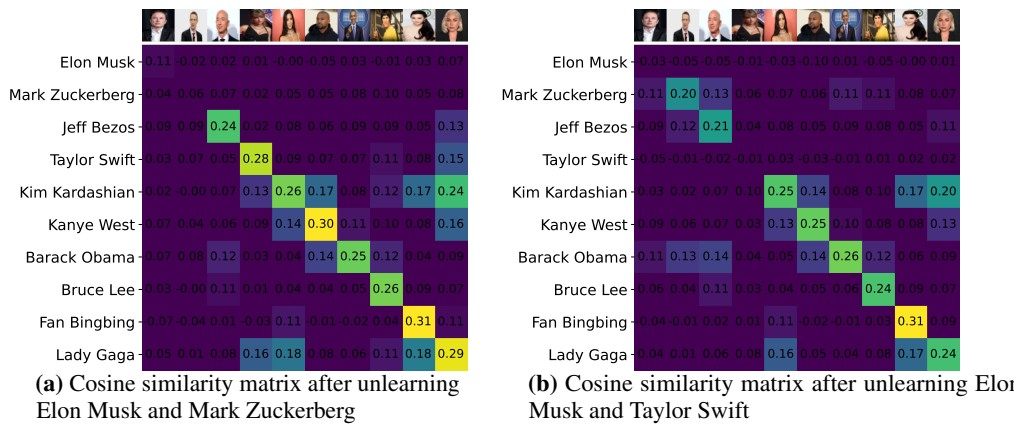

**(a)** Cosine similarity matrix after unlearning Elon Musk and Mark Zuckerberg      **(b)** Cosine similarity matrix after unlearning Elon Musk and Taylor Swift

**Figure 7:** Cosine similarity matrix of image-text pairs after unlearning multiple identities (see Figure 6a for the original model). (a) Unlearning Elon Musk and Mark Zuckerberg. (b) Unlearning Elon Musk and Taylor Swift. In both cases, the selected identities show disrupted alignment, while other identities remain largely unaffected. Based on the Pareto fronts in Figures 22a and 22e, we updated the vision layers `9.attn.out_proj` for Elon Musk and `11.attn.out_proj` for the second identity. Experiments were conducted on CLIP `ViT-B-32`.

We further analyze SLUG's performance as the number of identities to be unlearned increases. The identified layers are updated in parallel to achieve unlearning for $N$ identities. Figure 8 demonstrates effective unlearning across different values of $N$. The corresponding Pareto-front is detailed in Section G.

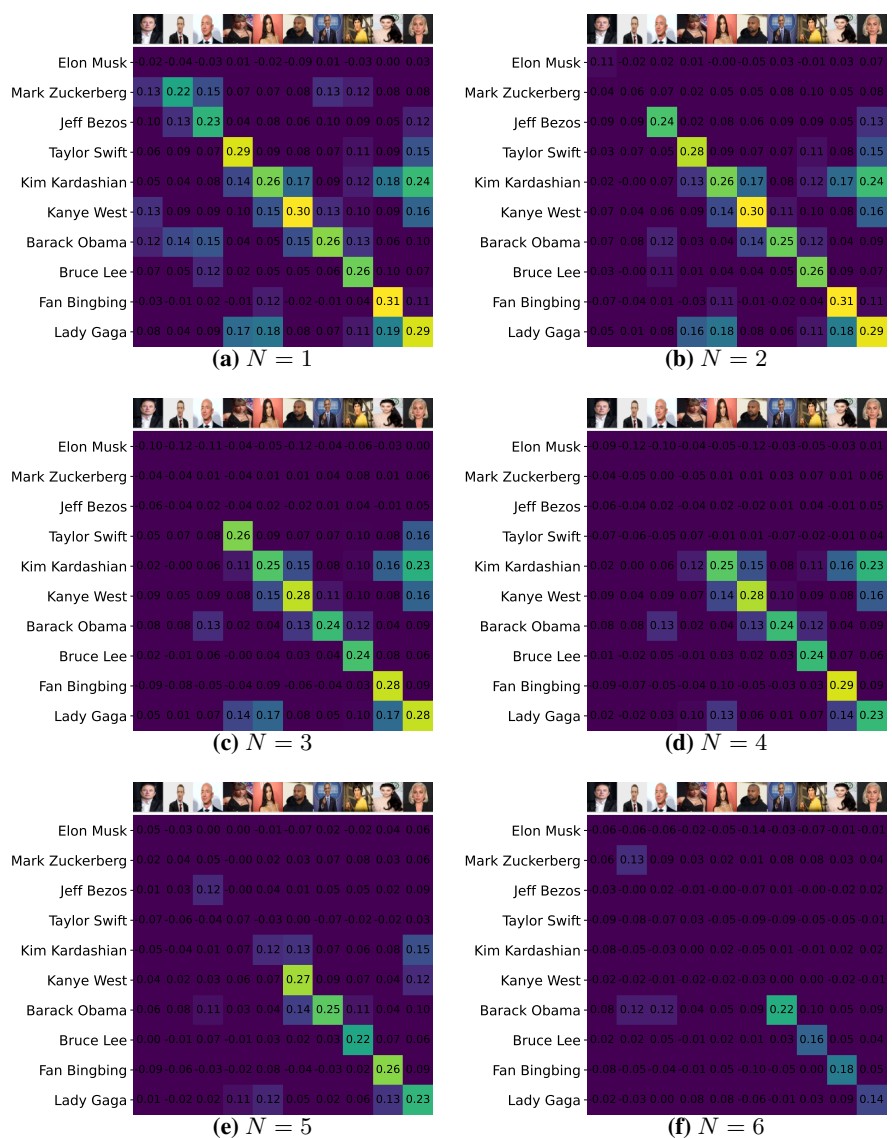

**Figure 8:** Cosine similarity matrices as we unlearn $N$ identities, where $N \in \{1, 2, ..., 6\}$. (a)–(f) Unlearn Elon Musk, Mark Zuckerberg, Jeff Bezos, Taylor Swift, Kim Kardashian, and Kanye West in a joint manner. To unlearn $N$ identities, our method (SLUG) identifies up to $N$ layers in the model using the single gradient calculated with the original network weights. The identified layers are then updated in parallel to achieve unlearning of $N$ identities.

## C.3 More CLIP architectures

We performed experiments using an expanded set of model architectures. The results for {`ViT-B-16` are discussed above in Figure 6. The results for `ViT-L-14, EVA01-g-14`} are discussed in Figures 9,10, respectively. These results demonstrate our method offers scalability and effectiveness across a range of model sizes, from 149.62 million parameters (`ViT-B-16`) to 1.136 billion parameters (`EVA01-g-14`). This underscores the flexibility of our approach to accommodate models of different scales. The Pareto-front of this experiment is included in Section G, where shows the metrics for different layers that our method uses to identify significant layers.

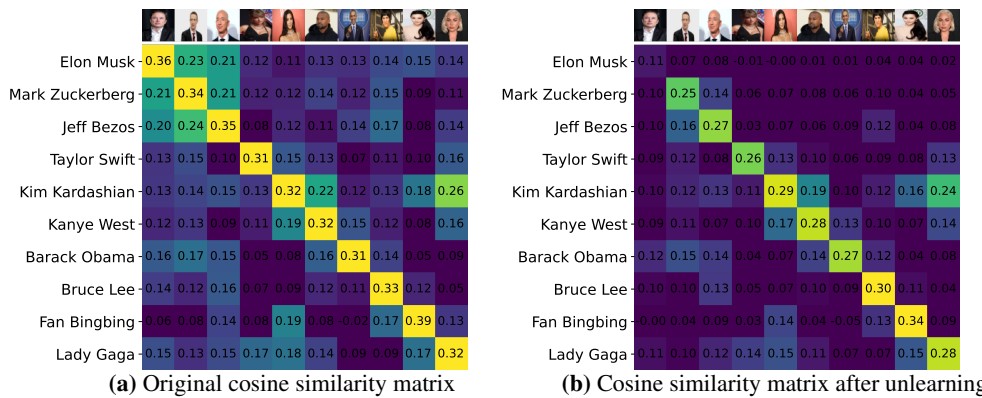

**(a)** Original cosine similarity matrix      **(b)** Cosine similarity matrix after unlearning

**Figure 9:** Cosine similarity matrix of image and text pairs before and after unlearning Elon Musk. After unlearning, the image and text pair of Elon Musk are not matched, while other persons are only slightly affected. Here, based on the pareto front in Fig. 23c, we select and update the vision layer `23.mlp.c_fc` for unlearning. CLIP model: `ViT-L-14`

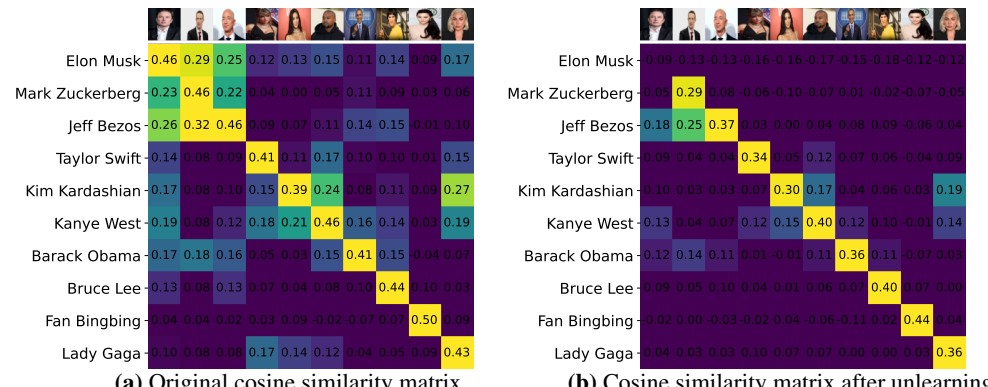

**(a)** Original cosine similarity matrix      **(b)** Cosine similarity matrix after unlearning

**Figure 10:** Cosine similarity matrix of image and text pairs before and after unlearning Elon Musk. After unlearning, the image and text pair of Elon Musk are not matched, while other persons are only affected. Here, based on the pareto front in Fig. 23f, we select and update the language layer `11.attn.out_proj` for unlearning. CLIP model: `EVA01-g-14`.

## C.4 Unlearning object concepts in CLIP

In addition to unlearning identities from CLIP, we also sample 7 classes {Basketball, Beach, Castle, Revolver, Rifle, School bus, Sunglasses} from ImageNet to evaluate the unlearning performance of our method on object concepts. For this experiment, we use 10k ImageNet validation images and sample images associated with target classes to create forget sets and compute gradients to unlearning different classes from the CLIP model. For evaluation, we use zero-shot accuracy reduction as the metric of effective unlearning target classes from the CLIP. The results, presented in Table. 6, show the CLIP zero-shot accuracy evaluations for both the forgetting of sampled classes and the retention of other ImageNet classes after unlearning. Our findings indicate that our method effectively reduces the CLIP zero-shot accuracy for the targeted classes to 0.0%, while the accuracy for remaining classes remains high, experiencing only minimal degradation (ranging from 0.03% to 2.03%) compared to the original pre-trained model, which indicates that the model's original functions are highly preserved after our unlearning.

**Table 6:** Unlearning performance of our method on common object concepts. FA@1 and FA@5 represents the top-1 and top-5 forget accuracy (%) of each forget class (i.e., zero-shot classification accuracy of unlearned class). TA@1 and TA@5 represents the top-1 and top-5 accuracy (%) of all classes of ImageNet except the corresponding Forget class. Each row shows the forget class accuracy and average accuracy over all classes of ImageNet before and after unlearning a class. Our method can reduce the forget accuracy of Forget classes to $0.0\%$ while keeping the accuracy of the remaining classes close to original model (within $0.06 - 2.03\%$ difference). CLIP model: `ViT-B-32`. TA@1 and TA@5 for the original model remains almost the same for all rows; therefore, we list it once in the table.

| FORGET CLASS | ORIGINAL | | | | UNLEARNED | | | |
|---|---|---|---|---|---|---|---|---|
| | FA@1 | FA@5 | TA@1 | TA@5 | FA@1 $\downarrow$ | FA@5 $\downarrow$ | TA@1 $\uparrow$ | TA@5 $\uparrow$ |
| BASKETBALL | 100.0 | 100.0 | | | 0.0 | 0.0 | 59.18 | 84.48 |
| BEACH | 54.55 | 72.73 | | | 0.0 | 0.0 | 59.54 | 84.78 |
| CASTLE | 87.50 | 100.0 | | | 0.0 | 0.0 | 58.13 | 83.87 |
| REVOLVER | 100.0 | 100.0 | 60.16 | 85.52 | 0.0 | 0.0 | 59.94 | 85.43 |
| RIFLE | 42.86 | 57.14 | | | 0.0 | 0.0 | 60.08 | 85.49 |
| SCHOOL BUS | 76.92 | 100.0 | | | 0.0 | 0.0 | 59.50 | 89.18 |
| SUNGLASSES | 44.44 | 55.56 | | | 0.0 | 0.0 | 60.13 | 85.23 |

### C.5  Impact of unlearning on semantically similar objects

Our method is designed to address precisely this concern by balancing unlearning effectiveness with utility preservation. We identify the most critical layer to update using layer importance and gradient alignment metrics that minimize impact on retained information while maximizing unlearning of targeted concepts. This approach allows for precise targeted removal while preserving general model performance on both related and unrelated tasks. Our experimental results demonstrate this balance. When unlearning specific identities in CLIP, our approach achieves state-of-the-art results while maintaining high accuracy on the CelebA dataset (containing many semantically similar identities) with only minimal degradation compared to the original model ($58.32\%$ vs. $61.38\%$ top-1 accuracy). This significantly outperforms other methods like SSD, which drops to $35.96\%$ accuracy. SLUG shows minimal impact on related concepts and image quality across all our experiments, demonstrating its effectiveness at avoiding over-unlearning of semantically similar objects.

To further quantify the impact of unlearning on semantically similar objects, we sampled the "Basketball," "Revolver," and "School Bus" rows from Table 6 and conducted additional zero-shot classification evaluations on the unlearned CLIP models. The semantically related classes were selected based on the ImageNet hierarchy and the top-5 most likely ranks in the logits across all targeted instances. The results in Table 7 indicate that the zero-shot accuracy of unlearned CLIP on both semantically related and top-5 most-likely classes remains high, comparable to its performance on the full ImageNet zero-shot evaluation. This further demonstrates the strong utility retention of our approach.

**Table 7:** Additional evaluation of unlearned models on classes that are semantically close to the forget class, and top-5 most-likely classes from the classification logit vectors. SLUG unlearned models maintain high test accuracy over classes that are closely related to the target.

| FORGET CLASS | FORGET ACC ($\downarrow$) | TEST ACC IMAGENET ($\uparrow$) | TEST ACC SEMANTICALLY RELATED CLASSES ($\uparrow$) | TEST ACC TOP-5 MOST-LIKELY SELECTED FROM LOGIT ($\uparrow$) |
|---|---|---|---|---|
| BASKETBALL | 0.0 | 59.18 | 73.63 | 58.33 |
| REVOLVER | 0.0 | 59.94 | 43.59 | 38.89 |
| SCHOOL BUS | 0.0 | 59.50 | 86.21 | 78.85 |

## C.6 Linearity of unlearning trajectory of different layers

In addition to the layers presented in Figure 2 (c) and (d), we show in Figure 11 that different layers show similar unlearning behaviors if we update them along their respective gradient direction (computed once for the original model). Nevertheless, the utility performance may vary depending on the selected layer; thus, it is important to select the best layer from the Pareto set for the overall best performance.

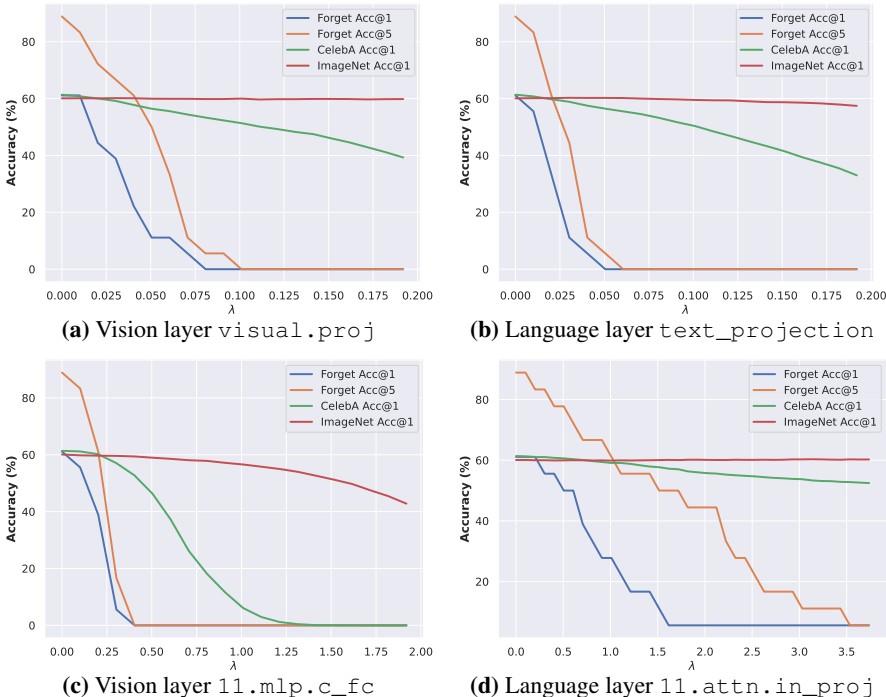

**(a)** Vision layer `visual.proj`

**(b)** Language layer `text_projection`

**(c)** Vision layer `11.mlp.c_fc`

**(d)** Language layer `11.attn.in_proj`

**Figure 11:** More examples of unlearning different layers. Correspond to Figure 2. The performance changes monotonically with the step size $\lambda$.

# Appendix D  More evaluations on unlearning Stable Diffusion

To demonstrate the performance and practical utility of our method, we further conduct a robustness study of SLUG in Section D.1, providing additional qualitative evaluation on scenarios in Section D.3 (e.g., more identities, copyright characters, novel concepts, artistic styles). Additionally, we provide experimental details for evaluating SLUG on *UnlearnCanvas* in Section D.5.

## D.1 Blackbox adversarial and quantization robustness

Recent research has exposed flaws in the robustness of foundation models unlearning. Notably, the Concept Arithmetic Attack (CRA) (Petsiuk & Saenko, 2025) demonstrates an optimization-free method where attackers exploit concept arithmetic properties of SD to reconstruct "unlearned" content through composite prompts. Zhang et al. (2025) show that loading post-unlearning LLM weights at lower-bit precision (higher quantization) significantly weakens the unlearning effect observed at higher-bit precision. These findings highlight how simple manipulations can undermine unlearning, raising serious concerns about the reliability of existing methods.

To verify effectiveness of SLUG, we applied both the Concept Arithmetic Attack and quantization to SD models unlearned by SLUG. Figure 12 shows that SLUG remains robust to concept arithmetic, as it modifies the encoder component, disrupting concept interoperability and effectively influencing downstream text-guided image generation. In Figure 13, we test an unlearned model trained in 16-bit floating point (fp16) by loading it in 8-bit unsigned integer (Uint8). The results demonstrate that SLUG maintains its unlearning effect despite quantization.

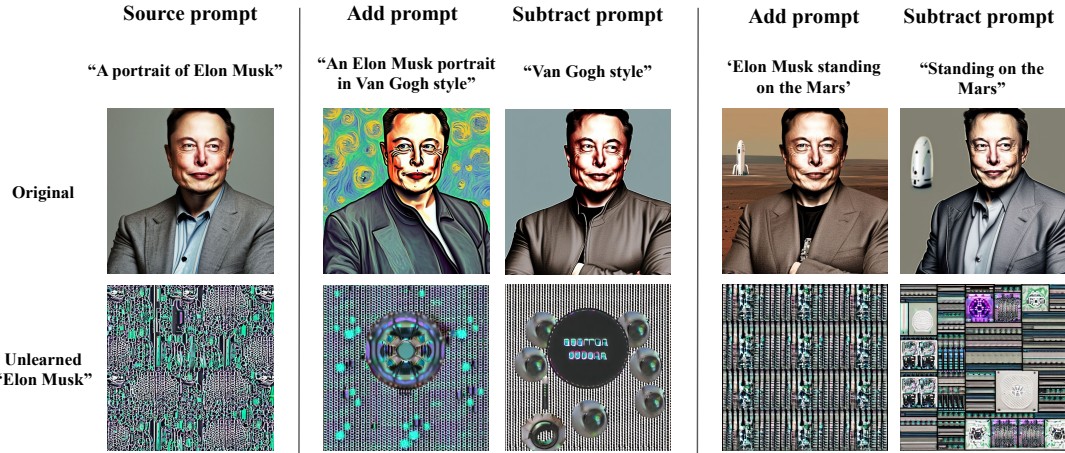

**Figure 12:** SLUG is robust to Concept Arithmetic Attacks. The first row illustrates the concept arithmetic property of Stable Diffusion, where distinct concept groups are added or subtracted from the source prompt. Each column presents the generated image using the corresponding arithmetic prompt. The model unlearned by SLUG fails to generate the target ID consistently across different arithmetized prompts.

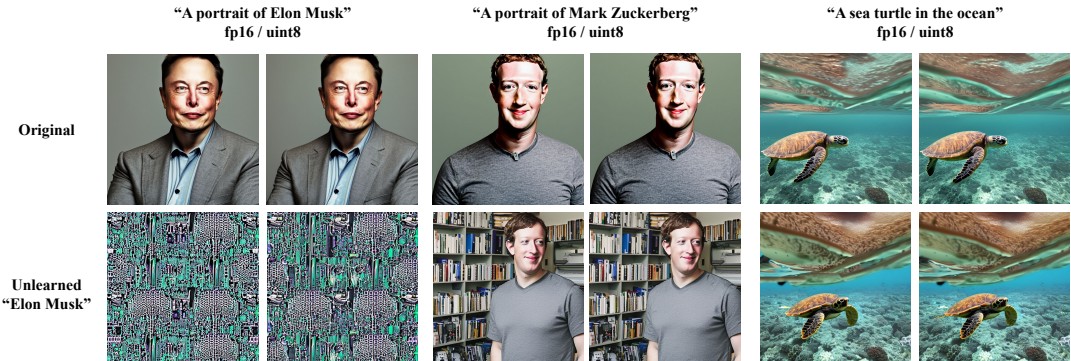

**Figure 13:** SLUG is robust to model weight quantization. The first row serves as a sanity check, showing that the original pretrained SD model generates images consistently across different quantization levels. Post-unlearning models with SLUG exhibit negligible differences, consistently failing to generate the targeted ID while preserving utility for other concepts.

### D.2 Whitebox adversarial robustness.

In Table 8, we utilize the latest UnlearnDiffAtk (Zhang et al., 2024e) and P4D (Chin et al., 2024). Specifically, we selected the "Nudity" and "Church" from Table 2 and Table 4 of Zhang et al. (2024e) to provide a brief adversarial evaluation of SLUG.

Following the same setup as Zhang et al. (2024e), we applied SLUG on SDv1.4 to unlearn "Nudity" concept and "Church" object, then attack the SLUG-unlearned SDv1.4 using two attack methods: UnlearnDiffAtk and P4D that optimized 142 and 50 adversarial prompts for "Nudity" and "Church," respectively.

Having robustness against whitebox attacks is challenging without corresponding adversarial design in the unlearning process. The results indicate that SLUG (like other unlearning methods) is not immune to whitebox adversarial attacks, yet SLUG demonstrates effectiveness on unlearning NSFW concepts and objects that are studied in existing literature.

### D.3 More unlearning scenarios

**More celebrity names.** Beyond unlearning "Elon Musk" from Stable Diffusion, which is presented in Figure 4, here we also provide additional qualitative evaluations on unlearning other celebrity names {Taylor Swift, Jeff Bezos}

**Table 8:** Evaluation against adversarial attacks. Lower ASR (%) indicates better adversarial robustness. Row-*No Attack* shows the original performance on unlearning tasks.

| CONCEPT/OBJECT | | NUDITY | | | | CHURCH | | |
|---|---|---|---|---|---|---|---|---|
| UNLEARN METHOD | | OURS | ESD | FMN | SLD | OURS | ESD | FMN |
| ATTACKS: (ASR %) | NO ATTACK | 16.90 | 20.42 | 88.03 | 33.10 | 4 | 14 | 52 |
| | P4D | 76.76 | 69.71 | 97.89 | 77.46 | 46 | 56 | 98 |
| | UNLEARNDIFFATK | 90.32 | 76.05 | 97.89 | 82.39 | 80 | 60 | 96 |

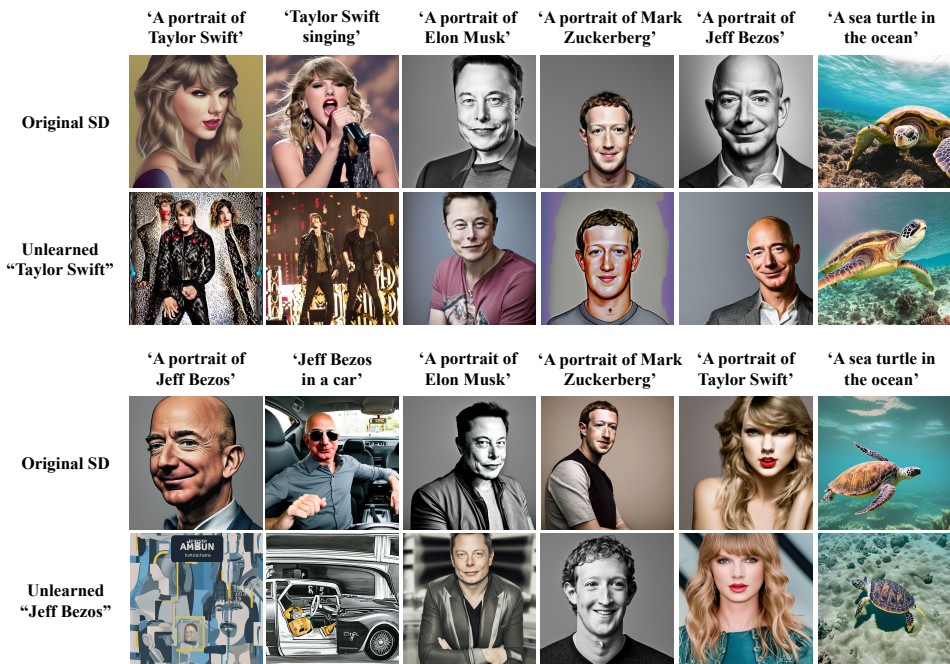

**Figure 14:** Qualitative evaluation on unlearning celebrity names Taylor Swift and Jeff Bezos from the Stable Diffusion.

with our method in Figure 14.

**Unlearning concepts and copyright contents.** In addition to identity removal for privacy protection, we address copyright concerns that increasingly challenge generative models. For unlearning copyrighted contents from Stable Diffusion models, we generate 500 images using unlearning targets as prompts, and use them as the forget set. The retain set is a single shard of LAION-400M dataset, same as for CLIP unlearning.

We successfully apply our method to remove copyright-protected content, specifically targeting well-known characters such as Marvel's "Iron Man" and Walt Disney's "Mickey Mouse." Figure 15 illustrates that our technique precisely unlearns the targeted concepts, effectively disabling the generation of images associated with these copyrighted entities while preserving the ability of the model to produce images of other concepts. These results demonstrate the use of SLUG in protecting intellectual property from generative AI.

**NSFW concepts.** Our experiments in Table 8 follow the setup of UnlearnDiffAttack (Zhang et al., 2024e), which considers more practical unlearning scenarios on NSFW (not-safe-for-work) concepts. We applied SLUG to unlearn the "Nudity" concept in SDv1.4. The results in NO ATTACK row of Table 8 demonstrate that SLUG is applicable to NSFW concepts previously studied in related work.

**Novel concepts.** One of the intriguing properties of the Stable Diffusion is its ability to generalize image generation to novel concepts that are infrequently or never observed in the real world. In this experiment, we explore the unlearning of a unique concept, "Avocado chair" from Stable Diffusion. We first generate 500 image using SD with the prompt "An avocado chair" to create the forget set, and use the same retain set as other experiments, which is is a single shard of LAION-400M dataset. In Figure 16, we show that our method successfully unlearn the concept "Avocado chair" from SD, resulting in the model's

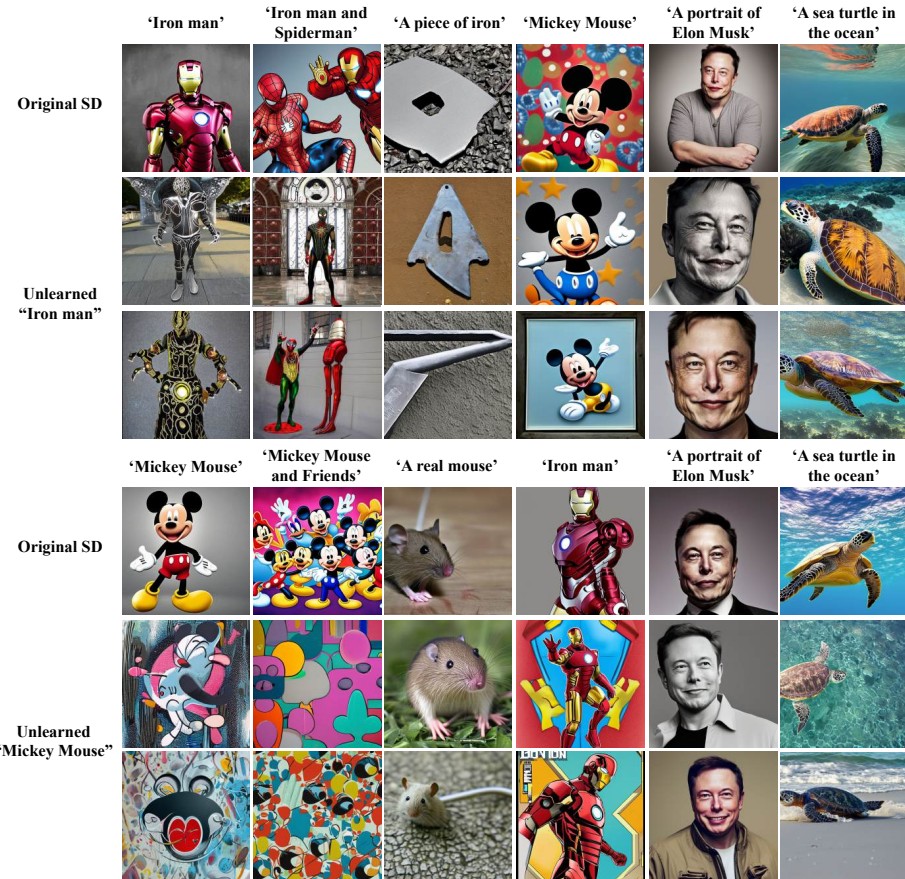

**Figure 15:** Qualitative evaluation of unlearning copyrighted characters "Iron Man" and "Mickey Mouse" from Stable Diffusion. The first row shows images from the original pretrained model, while the second and third rows display outputs from the unlearned model using the prompts above each column. Our method effectively removes copyrighted concepts while preserving overall image generation quality.

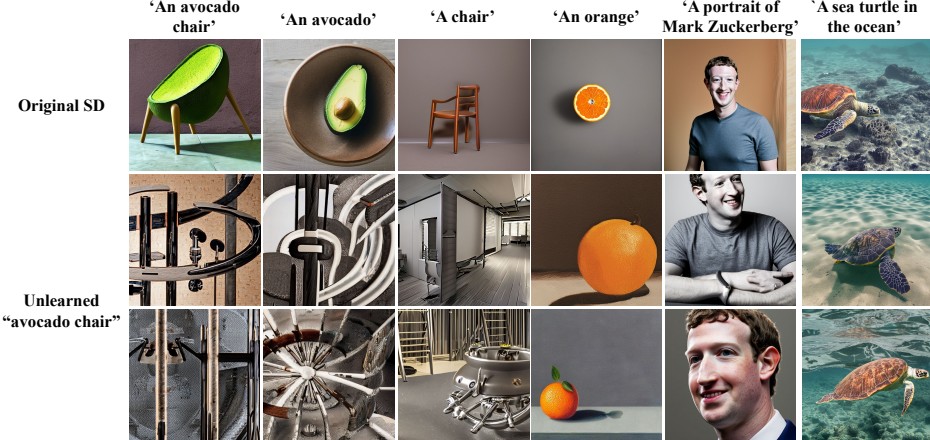

**Figure 16:** Qualitative evaluation on unlearning a novel concept "Avocado chair" from the SD.

inability to generate images corresponding to this specific concept.

It is noteworthy that the model's capability to generate images related to the constituent atomic concepts (namely "Avocado" and "Chair") is also compromised. We hypothesize that this occurs due to the model's treatment of novel concepts as compositions of atomic concepts. For example, the concept "Avocado chair" is interpreted by the model as "Avocado" plus

"Chair." Consequently, when a novel concept is unlearned, the associated atomic concepts are inadvertently affected as well. This highlights a challenge in the model's approach to handling the interoperability of novel and atomic concepts.

**Artistic styles and object.** In the experiment of evaluating SLUG performance on UnlearnCanvas benchmark discussed in Section. 3.3, we use 400 images that are associated with each style, as the forget set for unlearning style, and 1200 images that are associated with each object concept as the forget set for unlearning object, all images are from the benchmark dataset. We use a single shard of LAION-400M dataset as the retain set.

For qualitative evaluation of this experiment, we provide visual examples of unlearning artistic styles: {Pop Art, Crayon, Sketch, Van Gogh} and object: dog that are sampled from UnlearnCanvas, in Figure 17, 18 and 19. These results further show the effectiveness of SLUG in unlearning a broad spectrum of concepts ranging from concrete (e.g., celebrity name, intellectual property figure, and object) to abstract (e.g., novel concept and artistic style).

## D.4 Details on Gap Ratio evaluation

In this section, we provide additional analysis of the Gap Ratio (GR) evaluation in Table 2, by breaking it down in terms of effectiveness and efficiency aspects in Table 9. We follow the same setup as Table 2, where we represent metrics for each method as a 9-dimensional GR vector, quantifying its gap from the "Best (hypothetical reference) method". To provide a more comprehensive geometric characterization of the performance gap, we report the $\ell_1$ and $\ell_2$ distance of the GR vectors, both normalized by the vector length (in this case $\ell_1$ is equivalent to average), in the ALL METRIC column of Table 9.

We also report summary metric using only effectiveness or efficiency metrics. In the EFFECTIVENESS column, we compute the distances using only the first 7 entries in the GR-vectors (corresponding to UA, IRA, CRA for both style and object, and FID) and their counterparts in the BEST (hypothetical) method of Table 2. In the EFFICIENCY column, we compute the distance using only the last 2 entries in the GR-vectors (Time and the "sum of memory and storage") and their counterparts in the Best (hypothetical) method.

In summary, SLUG offers the best performance across the combined Effectiveness and Efficiency metrics in Table 2. While SLUG provides the best results in terms of Efficiency, its Effectiveness is among the top-performing methods. We also acknowledge that different choices of norms and weighting of individual metrics can provide us different results for the summary metric.

**Table 9:** Gap Ratio summary of different unlearning methods over metrics of Table 2. Low value means smaller performance gap from the best performing method, the best performance is highlighted. SLUG offers competitive performance in effectiveness metrics.

| METHOD | EFFECTIVENESS | | EFFICIENCY | | ALL METRICS | |
|---|---|---|---|---|---|---|
| | $\ell_1$ NORM | $\ell_2$ NORM | $\ell_1$ NORM | $\ell_2$ NORM | $\ell_1$ NORM | $\ell_2$ NORM |
| ESD | 0.20 | 0.11 | 81.04 | 78.55 | 18.17 | 17.46 |
| FMN | 0.47 | 0.24 | 6.51 | 4.72 | 1.81 | 1.07 |
| UCE | 0.64 | 0.37 | 5.50 | 5.08 | 1.72 | 1.17 |
| CA | 0.17 | 0.09 | 10.37 | 9.03 | 2.44 | 2.01 |
| SALUN | 0.06 | 0.03 | 12.32 | 9.11 | 2.78 | 2.03 |
| SEOT | 0.24 | 0.13 | 1.22 | 0.88 | 0.46 | 0.22 |
| SPM | 0.16 | 0.07 | 380.71 | 380.27 | 84.73 | 84.50 |
| EDIFF | 0.20 | 0.11 | 23.45 | 19.97 | 5.37 | 4.44 |
| SHS | 0.37 | 0.20 | 19.50 | 15.78 | 4.62 | 3.51 |
| SLUG (OURS) | 0.19 | 0.08 | 0.00 | 0.00 | 0.15 | 0.06 |

## D.5 Experiment details on UnlearnCanvas

**Models.** UnlearnCanvas targets unlearning styles and objects from an SDv1.5 model fine-tuned to generate 20 different objects in 60 distinct styles. The benchmark provides pre-trained SDv1.5 models for evaluation in Diffusers and CompVis implementations. In our experiment, correspondly, we focus on the CLIP text encoder used in SDv1.5 Diffusers implementation: openai/clip-vit-large-patch14 from HuggingFace.

**Computational time, memory, and storage.** The gradient computational time and memory usage of SLUG depends on several factors: computing resource, batch size, and size of the forget set. Note that while the details of the evaluation of efficiency metrics are not well defined in the original UnlearnCanvas, in Table. 2 we are reporting the best performance of SLUG can achieve on our computing resource NVIDIA A100 40GB. Specifically, the batch size is set to 1 for recording

the memory usage of SLUG, and to $16$ for recording its computational time. This batch size of $16$, is consistent with the sizes used in our other experiments. For SLUG storage consumption, as our method only requires storing the gradient values of a few layers on the Pareto front, the actual storage consumption is $43$ MB ($0.043$ GB), which by approximation is $0.0$ GB in the benchmark scale.

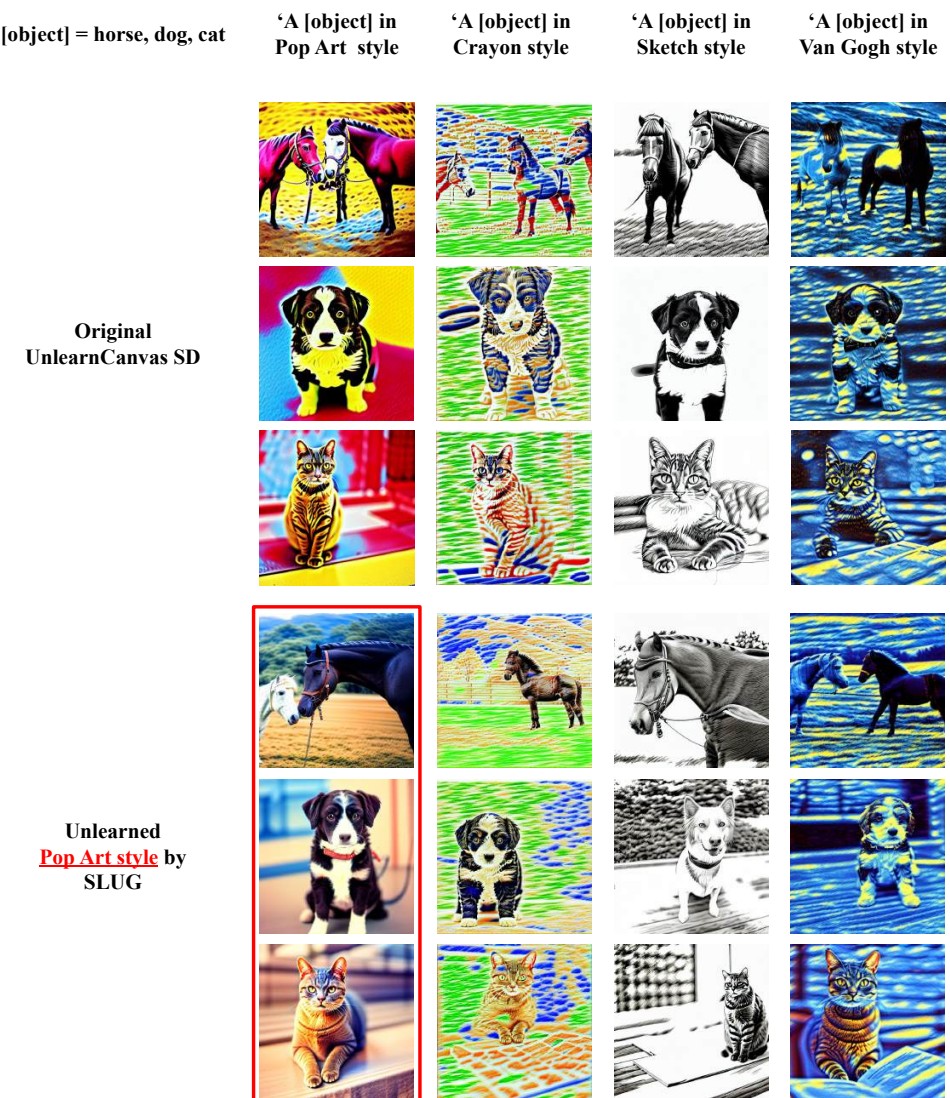

**Figure 17:** Visual examples of SLUG performance on UnlearnCanvas. Row $1 - 3$: outputs from original UnlearnCanvas Stable Diffusion (SD) using column captions as prompts. Row $4 - 6$: outputs from UnlearnCanvas SD unlearned Pop Art style. Outputs corresponding to the unlearned style are highlighted by the  red bounding box .

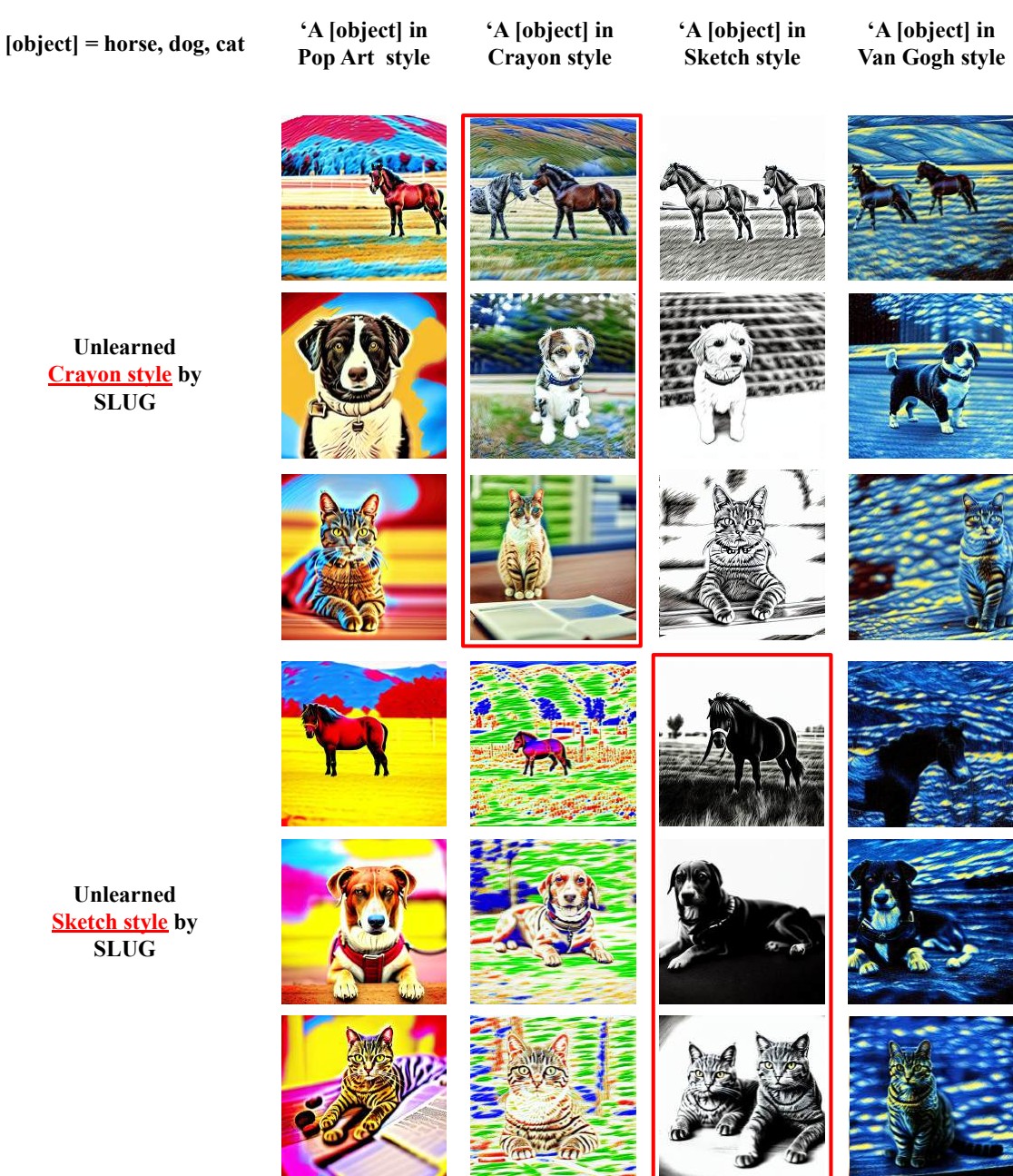

**Figure 18:** Visual examples of SLUG performance on UnlearnCanvas. Row $1 - 3$: outputs from UnlearnCanvas SD unlearned Crayon style. Row $4 - 6$: outputs from UnlearnCanvas SD unlearned Sketch style. Outputs corresponding to the unlearned style are highlighted by the red bounding box.

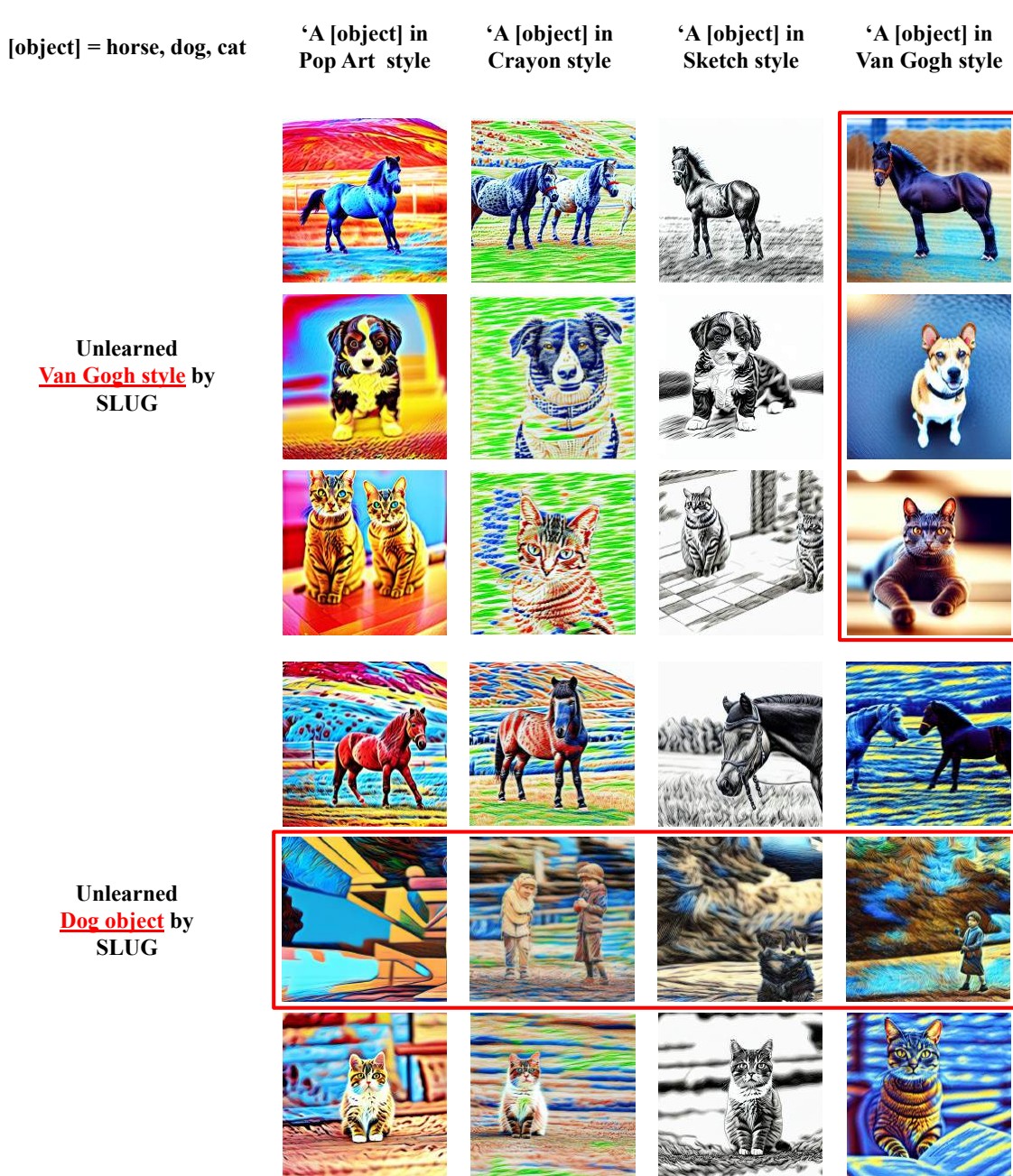

**Figure 19:** Visual examples of SLUG performance on UnlearnCanvas. Row $1-3$: outputs from UnlearnCanvas SD unlearned Van Gogh style. Row $4-6$: outputs from UnlearnCanvas SD unlearned dog object. Outputs corresponding to the unlearned style/object are highlighted by the red bounding box.

# Appendix E    More evaluations on unlearning VLM

In this section, we present additional qualitative examples of post-unlearning VLM with SLUG. Figure 20 showcases further responses from the unlearned LLaVA-1.5-7B model for the target identity "Elon Musk", beyond examples in Figure 5 in the main text.

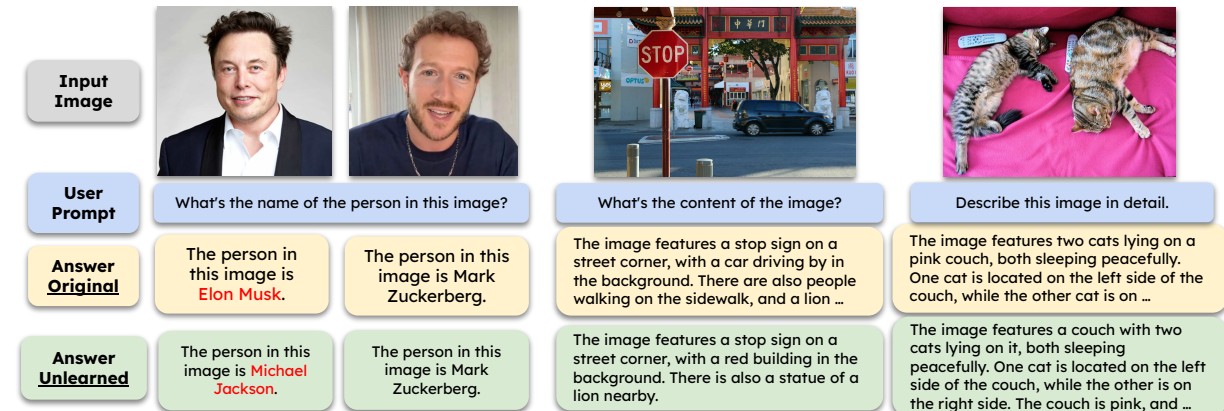

**Figure 20:** Qualitative evaluation on unlearning name "Taylor Swift" from LLaVA 1.5. While "Taylor Swift" is mapped to "woman" after the unlearning, the other female celebrity identification remain unaffected. Besides, model's robustness against style distribution shift is also preserved.

In addition to results presented in Section 3.4, we include more qualitative examples on unlearning a different identity "Taylor Swift" from LLaVA-1.5 in Figure 21. We demonstrate that our method can anonymize celebrity names from the pretrained Vision-language models, and simultaneously preserve the model's ability on image understanding, reasoning and distribution shift robustness on art work, cartoon style images.

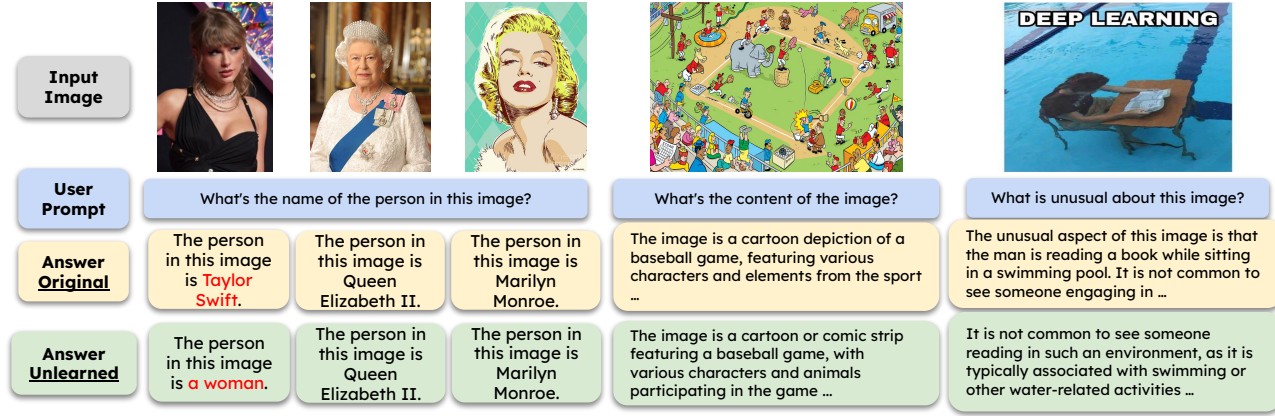

**Figure 21:** Qualitative evaluation on unlearning name "Taylor Swift" from LLaVA 1.5. While "Taylor Swift" is mapped to "woman" after the unlearning, the other female celebrity identification remain unaffected. Besides, model's robustness against style distribution shift is also preserved.

## Appendix F   Summary of model sizes

Our empirical results provide strong evidence supporting the claim of updating a single critical layer is sufficient/scalable for larger models. We conducted experiments across model scales summarized in Table 10, consistently demonstrating effective unlearning through single-layer updates (see Appendix C.3, Figure 9 and 10).

**Table 10:** Summary of total and unlearning manipulated parameter size of experimental models. Our approach is scalable to the largest model size up to 7B parameters.

| MODELS | TOTAL PARAMS | SINGLE LAYER PARAMS |
|---|---|---|
| CLIP VIT-B-32 | 151.28 M | 1.05 M |
| CLIP VIT-L-14 | 427.62 M | 2.36 M |
| CLIP EVA01-G-14 | 1.14 B | 2.35 M |
| SDv1.5, v2.1 | 983 M | 2.36 M |
| LLAVA 1.5-7B | 7.06B | 4.19 M |

## Appendix G   Pareto-fronts of all experiments

In this section, we present the complete set of Pareto-front plots for all the CLIP unlearning experiments discussed in Sections C.1, C.2, and C.3. These plots serve as a reference for our method, showing the identified layers for unlearning in each experiment.

**Sectioin C.1 More examples on unlearning identities:** Figure 22 illustrates the Pareto-front plots that are used to identify important layers selected by our method for unlearning different identities.

**Sectioin C.2 Joint update for unlearning multiple identities:** Figure 22 presents details on identifying layers associated with different identities and updating them to achieve unlearning of multiple identities at once.

**Sectioin C.3 More CLIP architectures:** Figure 23 shows the metrics for different layers that our method uses to identify significant layers for unlearning different CLIP architectures.

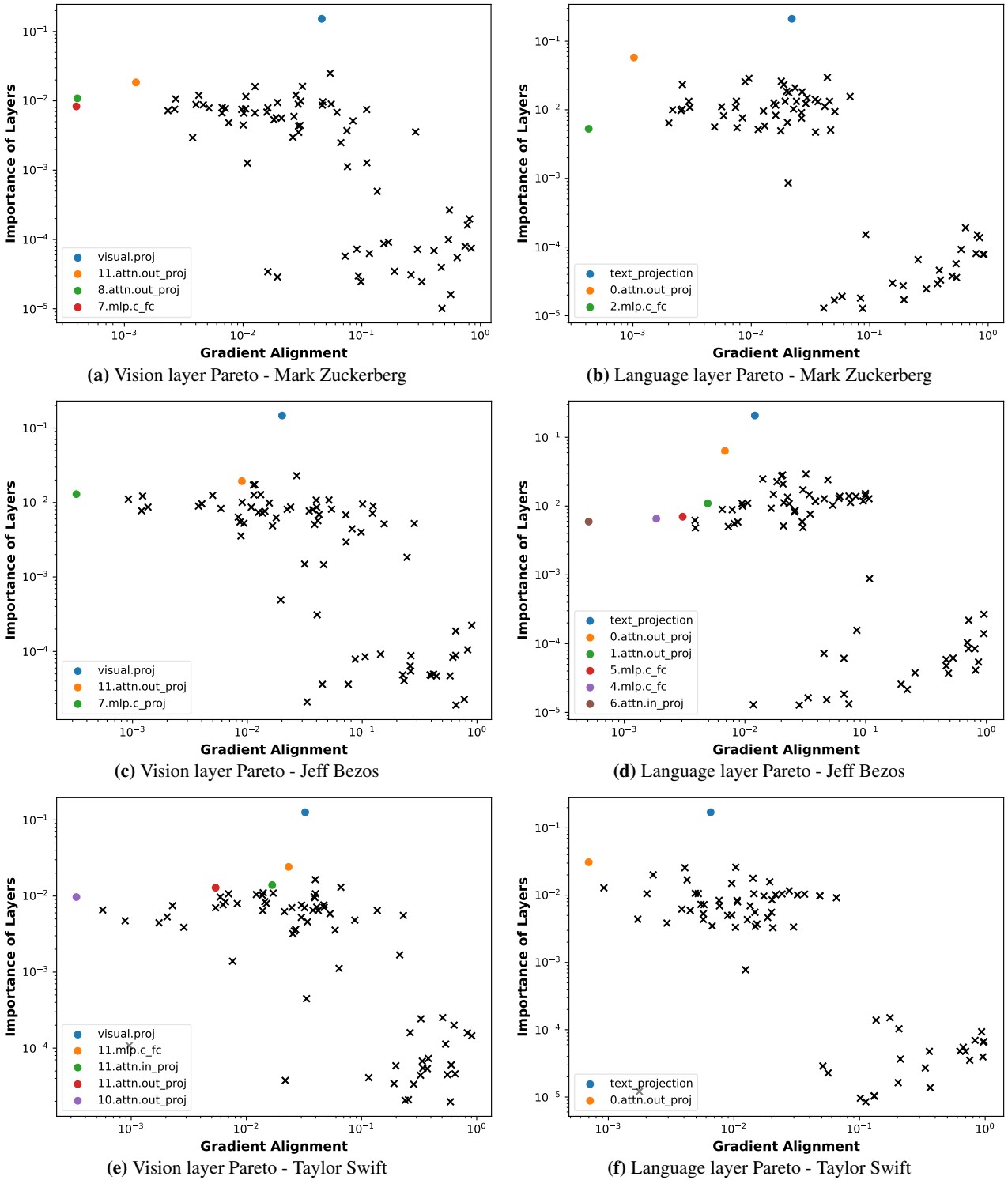

(a) Vision layer Pareto - Mark Zuckerberg

(b) Language layer Pareto - Mark Zuckerberg

(c) Vision layer Pareto - Jeff Bezos

(d) Language layer Pareto - Jeff Bezos

(e) Vision layer Pareto - Taylor Swift

(f) Language layer Pareto - Taylor Swift

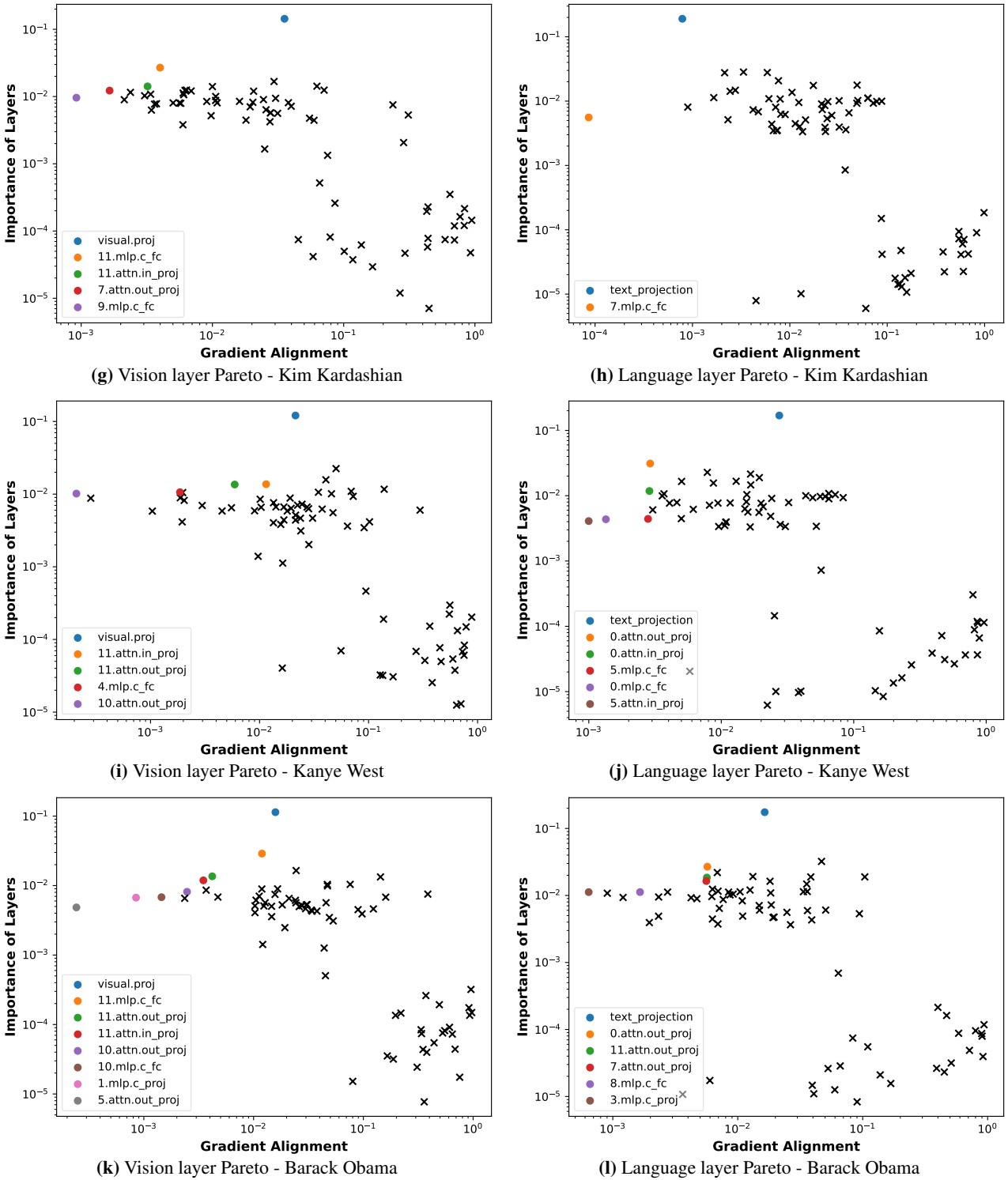

(g) Vision layer Pareto - Kim Kardashian

(h) Language layer Pareto - Kim Kardashian

(i) Vision layer Pareto - Kanye West

(j) Language layer Pareto - Kanye West

(k) Vision layer Pareto - Barack Obama

(l) Language layer Pareto - Barack Obama

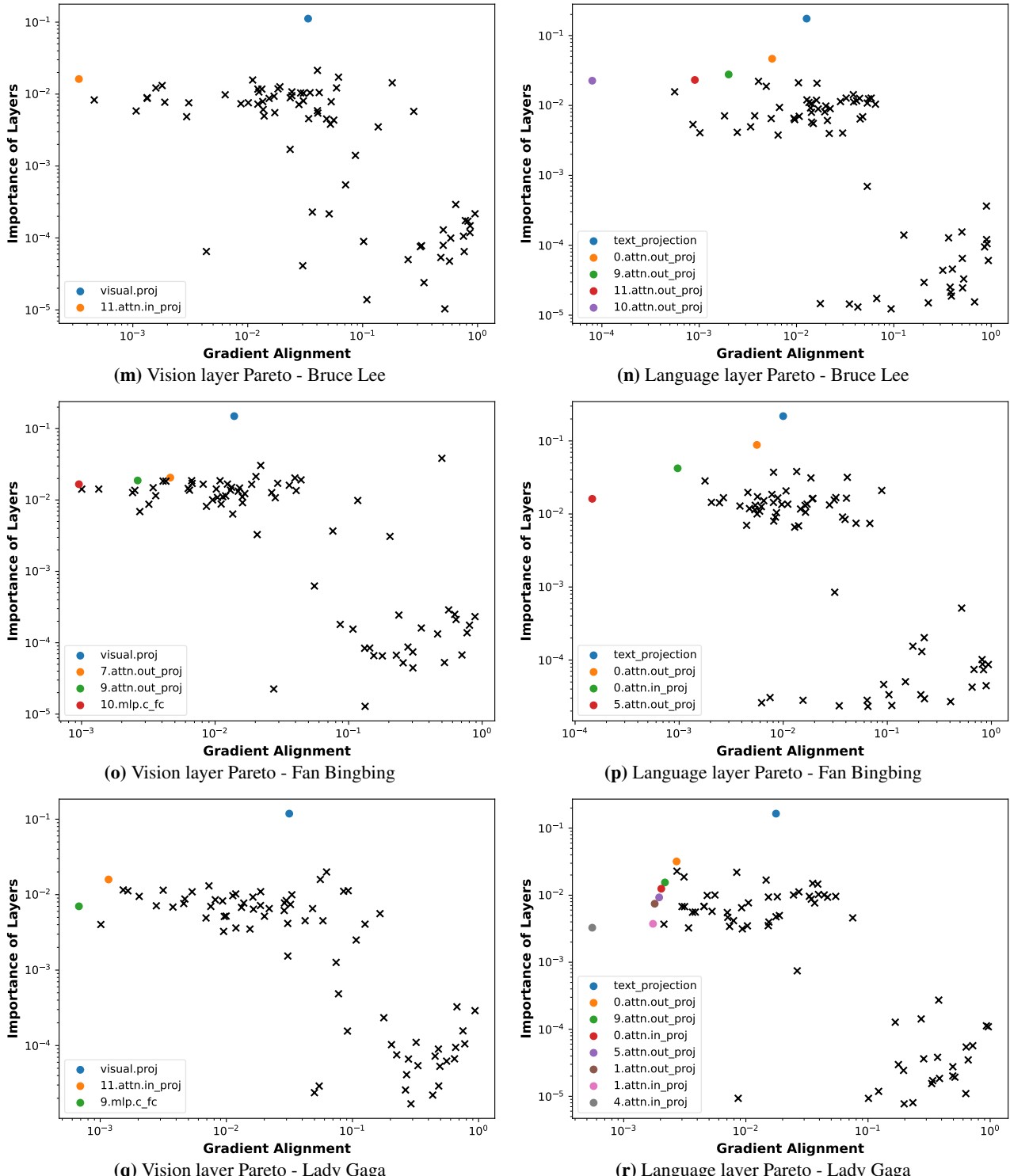

**Figure 22:** Scatter plots of layers for unlearning more identities, same setting as Figure 2. CLIP model `ViT-B-32`. Figures (a) - (r) shows the importance and gradient alignment of different vision model and language model layers as we unlearn different identities.

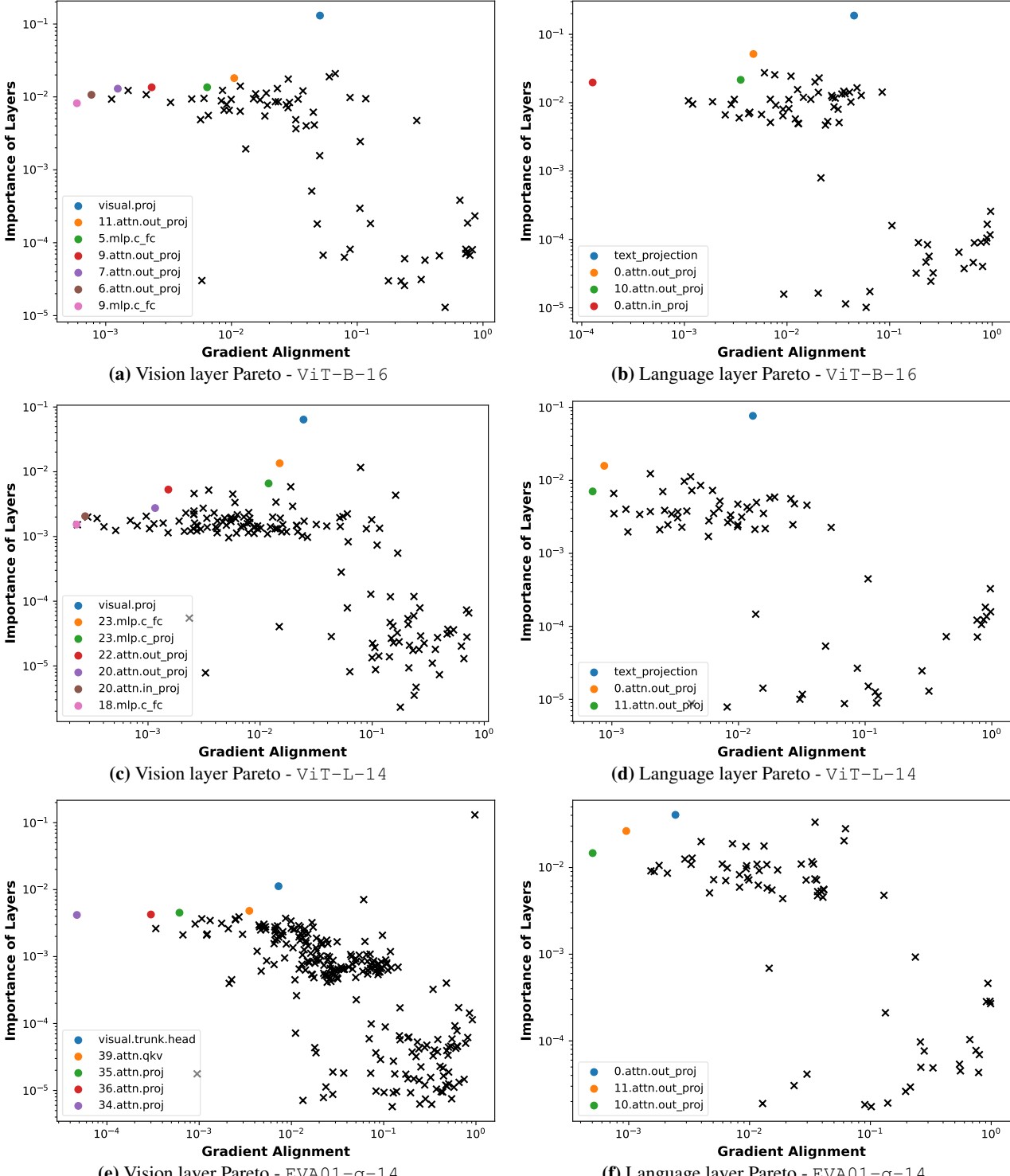

**Figure 23:** More CLIP models, in addition to Sec 3.2. Unlearning name `Elon Musk` from different CLIP models built in: {`ViT-B-16,` `ViT-L-14, and EVA01-g-14`}

