# OpenReview forum: "Targeted Unlearning with Single Layer Unlearning Gradient"
_ICML.cc/2025/Conference — ICML 2025 poster_

### Official Review · Reviewer_BJg7 · 2025-03-13

**Overall Recommendation:** 3

**Summary:**

This paper addresses the computational challenge and performance degradation often associated with machine unlearning methods, proposing an efficient technique called Single Layer Unlearning Gradient (SLUG). Instead of extensive updates across the entire model, SLUG strategically updates only a single critical layer, identified through metrics such as layer importance and gradient alignment. Experiments conducted on CLIP, Stable Diffusion, and vision-language models demonstrate SLUG’s effectiveness in unlearning both concrete elements (like specific identities or objects) and abstract concepts (such as artistic styles). Evaluations on the UnlearnCanvas benchmark indicate that SLUG achieves comparable unlearning accuracy to state-of-the-art methods, while significantly reducing computational cost. Thus, SLUG offers a practical, targeted unlearning approach that maintains model utility with minimal overhead.

**Claims And Evidence:**

Satisfactory.

**Essential References Not Discussed:**

The paper effectively covers recent and essential related approaches. However, it would be beneficial to also include text-to-image retrieval methods to demonstrate the performance of the CLIP model after unlearning target objects. Including such results and references would offer stronger and more comprehensive empirical evidence.

**Experimental Designs Or Analyses:**

1- This paper presents experiments focusing on the unlearning of target objects; however, it does not investigate the potential impact of this unlearning process on closely related objects. There is a concern that excessive unlearning (over-unlearning) could unintentionally affect semantically similar objects, which ideally should be retained. Therefore, further experiments are required to demonstrate clearly that the proposed approach avoids over-unlearning in classes closely related—but not identical—to the target objects.

2- For object/style unlearning in the CLIP model, experiments should also include evaluations on text-to-image retrieval tasks, as these can provide stronger and more compelling evidence of effective unlearning within the CLIP framework.

3- This paper primarily focuses on unlearning a single object, such as "Elon Musk," from a text-to-image diffusion-based generative model. After unlearning, the generated images currently result in random noise instead of depicting Elon Musk. For stronger and more convincing evidence, experiments should include scenarios involving complex textual prompts containing multiple objects alongside the unlearned object. The ideal outcome would be generating realistic images that accurately depict all other objects from the prompt while naturally omitting the target (unlearned) object, rather than producing random or noisy images. Including such observations would significantly strengthen the paper's empirical validation.

4-  A more intriguing experimental setup for style unlearning could be demonstrated as follows: suppose we aim to unlearn the "sketch" style from a text-to-image generative model. After unlearning, prompts explicitly requesting the sketch style—such as "a sketch of a dog" or "a sketch of a cat"—should fail to generate sketch-style images of these animals. However, if prompts requesting other styles—such as "a photo of a dog" or "a cartoon of a dog"—are provided, the model should correctly produce images consistent with those styles. Additionally, when using a neutral prompt like "generate a dog" or "generate a cat" without specifying any style, the resulting set of images should exclude any sketch-like representations. Conducting and presenting such observations would offer compelling evidence and significantly strengthen the findings on style-specific unlearning.

**Methods And Evaluation Criteria:**

This paper provides a detailed description of the proposed approach, along with extensive experimental results on multiple benchmark datasets, such as UnlearnCanvas, to validate the effectiveness of the method. Additionally, a comprehensive ablation analysis is presented.

**Other Comments Or Suggestions:**

Would applying unlearning to multiple layers yield more effective results?

**Other Strengths And Weaknesses:**

Strengths:
1- This paper proposes a computationally efficient approach for target object unlearning by applying the unlearning process to only a single layer.

2- The paper applies unlearning techniques to diffusion-based generative models and the CLIP model, conducting comprehensive experiments on both object and style unlearning. Additionally, it evaluates the proposed approach using recent and more challenging datasets, such as UnlearnCanvas.

3- A detailed ablation analysis is included to support the claims presented in the paper.

Weaknesses:

1- A comparison demonstrating the computational efficiency of the proposed approach relative to existing methods should be included.

2- Experiments demonstrating the impact of unlearning target objects on semantically related retained objects should also be included. For example, in a fine-grained dataset, the top 5-10 closely related objects to the target (unlearned) object can be identified using similarity scores calculated from the retained set after unlearning. Subsequently, the generative capability of the model for these closely related objects should be evaluated. Additionally, for the CLIP model, conducting text-to-image retrieval tasks involving these closely related objects after target object unlearning would provide insightful observations and strengthen the evidence presented.

3- To provide stronger and more compelling evidence, the experiments should include scenarios with complex textual prompts involving multiple objects, including the object targeted for unlearning. Ideally, the generated images should realistically represent all other mentioned objects while naturally omitting the target (unlearned) object, rather than producing random or noisy outputs. Incorporating such experimental observations would greatly enhance the empirical validation of the paper's claims.

**Questions For Authors:**

Please see the concerns highlighted in the Weaknesses and Experimental Designs or Analyses sections and address all.

**Relation To Broader Scientific Literature:**

The primary contribution of this work, compared to prior approaches, is the effective unlearning of the target object by applying the unlearning process only within a single layer. This strategy enhances both the effectiveness and computational efficiency. However, a direct comparison highlighting computational efficiency against previous approaches has not been included, which would further strengthen the claims and clearly demonstrate this advantage.

**Theoretical Claims:**

This paper primarily presents empirical results to support its claims; however, it does not provide any theoretical proofs.

---

> ### Author Rebuttal · Authors · 2025-04-01
>
> We thank the reviewer for their thoughtful feedback. Below, we address each point raised:
>
> ## Impact of unlearning on semantically similar objects
> Our method, SLUG, is designed to address precisely this concern by balancing unlearning effectiveness with utility preservation. We identify the most critical layer to update using layer importance and gradient alignment metrics that minimize impact on retained information while maximizing unlearning of targeted concepts. This approach allows for precise targeted removal while preserving general model performance on both related and unrelated tasks.
> Our experimental results demonstrate this balance. When unlearning specific identities in CLIP, our approach achieves state-of-the-art results while maintaining high accuracy on the CelebA dataset (containing many semantically similar identities) with only minimal degradation compared to the original model (58.32% vs. 61.38% top-1 accuracy). This significantly outperforms other methods like SSD, which drops to 35.96% accuracy.
>
> For precision, we show minimal impact on related concepts and image quality across all our experiments, demonstrating SLUG's effectiveness at avoiding over-unlearning of semantically similar objects.
>
> Below, we sampled the "Basketball," "Revolver," and "School Bus" rows from Table 4 and conducted additional zero-shot classification evaluations on these unlearned CLIP models. The semantically related classes were selected based on the [ImageNet hierarchy](https://observablehq.com/@mbostock/imagenet-hierarchy) and the top-5 most likely ranks in the logits across all targeted instances.
>
>
> The results indicate that the zero-shot accuracy of unlearned CLIP on both semantically related and top-5 most-likely classes remains high, comparable to its performance on the full ImageNet zero-shot evaluation. This further demonstrates the strong utility retention of our approach.
>
> Table R5: Additional evaluation of unlearned models on classes that are semantically close to the forget class, and top-5 most-likely classes from the classification logit vectors. SLUG unlearned models maintain high test accuracy over classes that are closely related to the target.
> | Forget  class | FA (↓)  | TA_IN(↑)  | TA_Semantic related (↑) | TA_Top-5 most-likely (↑) |
> |:---:|:---:|:---:|:---:|:---:|
> | Basketball | 0.0 | 59.18 | 73.63 | 58.33 |
> | Revolver | 0.0 | 59.94 | 43.59 | 38.89 |
> | School Bus | 0.0 | 59.50 | 86.21 | 78.85 |
>
> ## Text-to-image retrieval evaluations for CLIP
> We agree that text-to-image retrieval evaluation would strengthen our results. In our CLIP unlearning experiments, we evaluate zero-shot classification accuracy on both unlearned content (Forget Accuracy) and retained content (using ImageNet and CelebA). This provides a strong indication of CLIP's alignment between textual and visual representations.
>
> Following the standard zero-shot paradigm, predictions are based on the highest cosine similarity between image and text embeddings. Our comprehensive cosine similarity matrices (Figures 3, 6-10) effectively demonstrate the disruption of image-text alignment for unlearned content while preserving alignment for retained content, which directly addresses text-to-image retrieval performance.
>
> ## Complex prompt scenarios with multiple objects
> Our experiments focused on SLUG’s computational efficiency, precise unlearning, and minimal side effects.
>
> For Stable Diffusion, Figure 4 shows that after unlearning, our model successfully replaces an identity (e.g., "Elon Musk") with electronic circuits while preserving the "Mars" setting—unlike other methods that degrade image quality or affect non-targeted concepts.
>
> Appendix Figure 15 (“Iron Man and Spiderman”) further demonstrates retention of non-targeted objects post-unlearning. Figures 17-19 show that even after style unlearning, models still generate the correct objects in other styles.
>
> We appreciate the suggestion to test complex prompts with multiple objects alongside unlearned ones, and a more rigorous, quantifiable evaluation worth exploring further.
>
> ## Style unlearning setup
> Our UnlearnCanvas experiment comprehensively evaluates style unlearning on an SDv1.5 model fine-tuned for 20 objects in 60 styles.
>
> Figures 17-19 show that after unlearning a style (e.g., "Pop Art"), our method prevents its generation while preserving other styles. In Figure 18, while the unlearned model still produces black-and-white images for "Sketch style," classifiers identify them as "Ink Art," not the original Sketch style.
>
> We further assess our method using UnlearnCanvas metrics: UA (Unlearn Accuracy) for unlearning effectiveness, and IRA/CRA for utility retention.
>
>
> ## Computational efficiency comparison
> As addressed above, we provide a direct comparison of computational efficiency in Table 2 for the UnlearnCanvas benchmark.
>
> ## Applying unlearning to multiple layers
> Please refer to response to **Reviewer oTLd, Table R4**

---

> > ### Comment · Reviewer_BJg7 · 2025-04-04
> >
> > I appreciate the authors’ responses, as they addressed most of my concerns. However, I remain somewhat unconvinced regarding the experiments on top-5 related objects accuracy before and after unlearning. Overall, I believe the proposed approach is effective and has significant potential to advance research in unlearning for generative models. Considering the authors’ responses and the feedback from other reviewers, I lean toward accepting this paper, with a weak accept rating, and I recommend that the authors incorporate all the comments in the final camera-ready version.

---

> > > ### Author Response · Authors · 2025-04-05
> > >
> > > Thank you for acknowledging our rebuttal and supporting the paper for acceptance!
> > > We will certainly incorporate all the comments in the final version.
> > >
> > > In our top-5 related object experiment, we showed that while an object from the Forget Class is completely unlearned (i.e., 0 Forget Accuracy), the Test Accuracy on semantically-related and most-likely classes remains high. If you can kindly guide us on which aspect of this experiment was unconvincing, we will be happy to modify and expand the experiment in the revision.

---

### Official Review · Reviewer_C1HE · 2025-03-14

**Overall Recommendation:** 2

**Summary:**

This paper proposes Single Layer Unlearning Gradient (SLUG), a technique for targeted unlearning in large-scale multimodal models by updating only a single critical layer using one gradient computation. The authors demonstrate its efficiency for CLIP, Stable Diffusion, and Vision-Language Models, claiming robust removal of targeted concepts while preserving broader model utility. Their approach hinges on carefully identifying the most relevant layer and then modifying it along a single gradient direction.

**Claims And Evidence:**

The authors' claims are partially supported by their empirical results but require more thorough and concrete substantiation. Please see the details in the Other Strengths and Weaknesses.

**Essential References Not Discussed:**

The authors lack some papers related to their work. Please see the details in Other Strengths And Weaknesses.

**Experimental Designs Or Analyses:**

I have reviewed their experimental designs and analyses. Please see the details in the Other Strengths and Weaknesses.

**Methods And Evaluation Criteria:**

Yes, please see the details in the Other Strengths and Weaknesses.

**Other Comments Or Suggestions:**

Please see the details in Other Strengths And Weaknesses.

**Other Strengths And Weaknesses:**

Current unlearning techniques often require extensive model-wide updates and still fail to robustly forget targeted information. The proposed approach addresses these limitations by targeting a single layer for updates, providing a more computationally efficient way to remove unwanted target information. The authors emphasize a balance between efficient forgetting and retaining the model’s utility, which is critical for real-world, large-scale applications. The experiments show the great trade-off between efficiency and effectiveness.
However, the following questions need to be further justified.

1. Layer Selection Assumption
The paper claims that updating one critical layer is sufficient to remove unwanted content. However, this claim may not scale to large diffusion models or other foundation models. Could the authors provide any empirical or theoretical evidence of this claim? For example, even in language models, memorization occurs across different layers [1].

2. Single Gradient Direction
While the authors highlight that single gradient computation reduces the computational overhead, there is little evidence or discussion explaining why a single direction suffices or how closely this approximates a multi-step update in practice. For example, if this is the case, this approach should be able to achieve the best forgetting performance in Table 2. The authors show strong efficiency in UnlearnCanvas experiments, but the forgetting scores do not outperform baselines. For example, methods like SalUn or ESD show higher unlearning quality for style removal. Does the best trade-off here refer to considering the efficiency dimension? If so, this claim is quite debatable. While prior works have tried to address the trade-off between forgetting quality and model utility, under which justification we can compromise the effectiveness of unlearning for efficiency remains unclear to me. How do the authors assign weight between the two to determine the best trade-off?

3. Although prior work unlearns object concepts in diffusion models, could the authors clarify the real-world motivation for object unlearning beyond the fact that related studies do so? In particular, for diffusion models, I understand that there is copyrighted or harmful content learned during the training stage, which needs to be addressed properly for responsibility. However, in which perspective do we have to consider object unlearning for large-scale foundation models? For instance, concepts like cats and dogs represent public information readily available in open datasets. Should we consider this from a privacy perspective (e.g., individuals requesting erasure of their data) or a safety perspective?

4. The discussion of UnlearnCanvas could be improved with more specifics on how the forget and retain sets were formed. For instance, the authors mention one example format of the forget data: "A [object name] in [style name] style". In this case, do we focus on erasing both information? If so, what does the retained set look like?
Specifically, if a target style like "Da Vinci" is removed (one-side removal), it would help to see how other styles remain valid, such as "Monet".

5. I think it would be very beneficial if the authors could provide quantitative results of the robustness dimension with respect to relearning attacks and adversarial prompting [2,3]. This would better substantiate the claim that updating a single layer is sufficiently robust and that the forgotten knowledge remains inaccessible under adversarial conditions.

[1] Demystifying Verbatim Memorization in Large Language Models

[2] JOGGING THE MEMORY OF UNLEARNED LLMS THROUGH TARGETED RELEARNING ATTACKS

[3] DO UNLEARNING METHODS REMOVE INFORMATION FROM LANGUAGE MODEL WEIGHTS?

**Questions For Authors:**

Please see the details in Other Strengths And Weaknesses.

**Relation To Broader Scientific Literature:**

This work tackles important and crucial problems in unlearning, such as addressing the trade-off by finding a single layer and computing the single gradient direction. The results show a great balance among the three factors, which could potentially help the unlearning community fo future research.

**Theoretical Claims:**

This paper does not have a theoretical claim.

---

> ### Author Rebuttal · Authors · 2025-04-01
>
> We sincerely thank the reviewer for their thoughtful comments and constructive feedback on our paper. We address each of the concerns below:
> ## Layer Selection Assumption
> The reviewer questions whether updating a single critical layer is sufficient/scalable for larger models. Our empirical results provide strong evidence supporting this claim. We conducted experiments across model scales summarized in following table, consistently demonstrating effective unlearning through single-layer updates (Sec C.3, Figure 9-10).
> | Models | Total Params | Single layer Params |
> |:---:|:---:|:---:|
> | CLIP ViT-B-32 | 151.28 M | 1.05 M |
> | CLIP ViT-L-14 | 427.62 M | 2.36 M |
> | CLIP EVA01-g-14 | 1.14 B | 2.35 M |
> | SDv1.5, v2.1 | 983 M | 2.36 M |
> | LLaVA 1.5-7B | 7.06B | 4.19 M |
>
> Regarding the concern about memorization across different layers, our method specifically addresses this through the layer identification procedure (Sec 2.2), which evaluates all layers to find the one most critical for the target concept. Our approach fundamentally differs from LLM memorization studies because:
> - We focus on multi-modal foundation models where concepts are represented differently than in pure language models
> - Our layer importance metric and gradient alignment analysis (Equations 5-7) precisely identify which layer contains the most salient information about the target concept
> - Our experiments across different architectures (ViT-B/L, EVA), different task models (SD, VLM) demonstrate the generalizability of this approach
>
> ## Single Gradient Direction
> The reviewer questions our claim about single gradient direction sufficiency. We provide clear evidence in Figure 2 (b,e) and Figure 11, showing that performance changes monotonically with step size along a single gradient direction. Critically, Figure 2 (c,f) demonstrates that iterative methods (GA and GAFT) provide no advantage over our single-gradient approach while requiring substantially more computation.
> Regarding UnlearnCanvas results, our claim of "best trade-off" refers precisely to the balance between unlearning effectiveness, utility preservation, and computational efficiency. While some methods achieve marginally higher UA scores in Table 2, they require orders of magnitude more computation (e.g., SPM: 29700s vs. SLUG: 39s) and memory (e.g., SalUn: 30.8GB vs. SLUG: 3.61GB). Importantly, SLUG shows no significant underperformance in any metric (no red highlights), achieving consistently strong performance across all dimensions.
> ## Motivation for Object Unlearning
> While unlearning common objects may not seem directly impactful, it serves multiple purposes:
> - Research methodology: Provides a controlled setting to precisely evaluate unlearning techniques using clear metrics..
> - Model customization: Enables tailoring foundation models to exclude specific objects (e.g., restricting non-medical outputs in medical imaging).
> - Data governance: Lays the groundwork for extending unlearning to privacy-sensitive content.
> - Safety & responsibility: Demonstrates technical feasibility in controlling model outputs, a key step toward responsible AI.
>
> ## UnlearnCanvas Dataset Details
> The UnlearnCanvas benchmark considers 20 different object classes, each represented in 60 distinct styles, resulting in a total of 20×60=1200 combinations. For example, consider the combination of "dog" and "Van Gogh." The benchmark uses its fine-tuned SDv1.5 model to generate 20 images with the prompt "A dog in Van Gogh style."
> To construct a complete dataset for the "dog" class, the benchmark fixes "dog" and iterates through the remaining 59 styles, generating a total of 20×60=1200 images. Similarly, to create a complete dataset for the "Van Gogh" style, it fixes "Van Gogh" and iterates through the remaining 19 objects, resulting in 20×20=400 images.
> The benchmark focuses on unlearning either a single class or a single style at a time, ultimately producing 20+60=80 unlearned models for each unlearning method. When unlearning a single style, we use the 400 images available in that style as the forget set and the remaining dataset as the retain set. Similarly, when unlearning a single object, we use the 1200 images available of that object as the forget set and the rest of the dataset as the retain set.
> The reported metric values in Table 2 are averaged over all 80 unlearned models. We would further clarify the setup of this benchmark in the revision.
> ## Adversarial Robustness Evaluation
> We thank the reviewer for highlighting the latest adversarial robustness studies on LLMs [1,2]. While our primary focus is on vision-language foundation models, we acknowledge that [1,2] vulnerabilities may also apply to these models and are worth further research.
>
> Regarding adversarial prompting, we conducted a brief evaluation during the rebuttal phase, please refer to response to **Reviewer Ceck, Table R2**.

---

### Official Review · Reviewer_oTLd · 2025-03-14

**Overall Recommendation:** 3

**Summary:**

The authors propose a novel (saliency-based) unlearning method called SLUG that identifies a single layer in the model and performs only a single update step in this layer to minimize negative side-effects on the model’s utility. Compared to related works such as SalUn [1], they assign values to each layer based on how important they are for the unlearning and how aligned their gradients are with the utility loss, instead of computing a mask over all parameters based on the unlearning importance alone. Then, SLUG figures out a pareto-optimal subset of layers that are the most important but least aligned with the retain loss to minimize interferences with the utility of the model. Within this set, they search for the best single layer and the best step size (learning rate) to take for the single update using a proposed binary search approach. They instantiate their method for CLIP and evaluate it with CLIP-based models like Stable Diffusion (SD), vision-language models (VLMs), or CLIP itself on a variety of different unlearning scenarios.

[1] Fan, Chongyu, et al. "Salun: Empowering machine unlearning via gradient-based weight saliency in both image classification and generation." arXiv preprint arXiv:2310.12508 (2023).


## Final recommendation and justification

I appreciate the effort made to address my concerns and open questions. My original review was recommending a Weak Reject for this work due to concerns regarding some of the claims, specifically regarding SOTA results on UnlearnCanvas benchmark and SLUG’s adversarial robustness, some clarity issues with their method and contribution, and the lack of a discussion around the downsides of a single-layer approach. In the rebuttal, these concerns were mostly addressed through additional explanations and additional experiments, which is why I raised my rating from Weak Reject to Weak Accept, in the expectation that my points get addressed and that the missing discussion around the potential shortcomings of a minimal method like SLUG will be added to the paper. I share the other points made by other reviewers that the trade-off between the efficiency and the unlearning effectiveness, and the influence of the SLUG method on this is still a bit unclear, especially since the unlearning results of SLUG are consistently worse than, e.g., the ones achieved with SalUn (see Table R7). For that reason, I will stick to my rating of Weak Accept and not raise it further.

**Claims And Evidence:**

**SOTA results on UnlearnCanvas (?)**: The claim of the authors to achieve SOTA results and the best trade-off in Table 2 (UnlearnCanvas) appears to be not supported enough through quantitative evaluations. Appreciating that SLUG as a minimal method is performing well, this claim should be supported by some summarizing metric or visualization, e.g., some type of harmonic mean. UnlearnCanvas is a multi-facetted benchmark, where not a single method stands out as the best in all aspects.

**Robustness**: They claim that SLUG is robust to key flaws exposed in recent un-learning studies, i.e. Concept Arithmetic Attacks and quantization, but only show anecdotal evidence for that. They also do not even mention other key flaws like inversion-based attacks [5] for unlearning concepts from Stable Diffusion. It is fine that a minimally invasive method like SLUG is not robust in all regards, but the claims here should then be made more carefully.

[5] Pham, Minh, et al. "Circumventing concept erasure methods for text-to-image generative models." arXiv preprint arXiv:2308.01508 (2023).

**Essential References Not Discussed:**

Generally, the related work section appears to be too brief, especially w.r.t. SalUn [1] and SDD [2], which are the closest methods to the proposed SLUG. To understand the relations and differences better, it would help the reader to draw more explicit connections in the main paper.

[1] Fan, Chongyu, et al. "Salun: Empowering machine unlearning via gradient-based weight saliency in both image classification and generation." arXiv preprint arXiv:2310.12508 (2023).

[2] Foster, Jack, Stefan Schoepf, and Alexandra Brintrup. "Fast machine unlearning without retraining through selective synaptic dampening." Proceedings of the AAAI conference on artificial intelligence. Vol. 38. No. 11. 2024.

**Experimental Designs Or Analyses:**

Yes, I reviewed the experimental designs and already provided some concerns in Methods And Evaluation Criteria. Beyond that, I am generally confused by the role of their `eval` function in their algorithm that is used to obtain FA (forget accuracy) and TA (test accuracy) to identify the best layer and step size. They claim a single-layer single-step approach but invoking an inner evaluation method can come at a significant computation cost, especially for application scenarios like the VLM or SD. As a reader, it was not clear to me if the final test evaluation was used for that or if the method relied on a validation set to find those step sizes. If no validation set was used for this “inner” evaluation, the numerical results presented in this paper could be overfitted to the specific test set in all of those cases. Moreover, it is unclear if the invocation of the `eval` function is priced into the complexity estimate of your algorithm? As far as I understood, the evaluation within the algorithm does, e.g. for the identity unlearning with the VLM, compute the FA using the definition from lines 423-427, which sounds relatively compute-heavy. The SLUG method effectively evaluates different points in the parameter space and then takes the one that performs the best overall. This is also a form of multi-step training from my POV. A discussion around these questions in the main paper could help to clarify the matter for readers.

**Methods And Evaluation Criteria:**

Mostly yes, the authors evaluate their SLUG method in various scenarios with different types of CLIP-based models and thereby show the generality of their method. Unfortunately, they sometimes fail to paint a convincing picture with their evaluation, especially w.r.t. clarity around the choices of FA and TA in the different settings. Appreciating the breadth of their experiments to show a wide applicability of their method and understanding that in-depth evaluation in each of them might be out-of-scope, more clarity around what exactly has been done in each of these experiments will strengthen this work.

For example, according to Section 3.1, their identity erasure setup involved unlearning 100 different celebrities, which appears to be their major experimental setup but they do not describe the details of that in their results: Does Table 1 show the averages of unlearning one identity at a time or unlearning all of them in parallel?

In the object unlearning case, it is unclear why they chose their own custom ImageNet classes as opposed to using the ones that prior work already used to evaluate their methods. A quantitative comparison might have been simplified by that. For example, Table 2 in the SalUn [1] paper could have been a great starting point for such a comparison.

In the VLM identity unlearning case, a more in-domain metric akin to the employed forget accuracy is missing which measures the same for the other non-erased celebrities. The general VLM benchmarks are a good addition but might miss more in-domain forgetting related to other celebrities and thereby underestimate the loss in utility especially with the lack of comparison to any related method. It is also unclear what definition of the test accuracy is used when applying SLUG in this setting.

**Other Comments Or Suggestions:**

1. **Typo**: “Prato-optimal layers” → “Pareto-optimal layers” (in Line 312)

1. **Line plot colors**: The choice of colors in Fig. 2 is slightly confusing because they do not refer to the same things in the different subplots; consider changing that to make it easier for the reader to understand your helpful visualizations.

1. **Confusion matrix figures**: The “confusion matrix” visualizations (e.g., Fig. 3) help to understand but are also hard to read for the close-to-zero numbers and the darker colors. The small images of the celebrities on the x-axis are a playful addition but lower the quality of the plot when being obviously stretched along one of the dimensions to fit the 1:1 aspect ratio. Consider cropping a square out of the images when recreating your plot.

1. **Naming of hyperparameters**: Using `K` for the maximum number of steps in your binary search algorithm (Algo. 3) and then using `K` as well to denote the number of epochs in Table 1 can confuse the reader. Consider changing one of them.

1. **Single Layer Selection**: As a reader it was not fully clear in the end how the final single layer is selected. Looking at your pseudocode in Algorithm 1 helps but I think readers would appreciate a few more sentences about this part of your method in the main paper.

1. **GAFT**: You mention GAFT as a two-stage combination of Gradient Ascent (GA) + Finetuning (FT) but never really elaborate how you go about that, i.e. when having GAFT with 2 iterations, does that mean 2 GA iterations followed by 2 FT iterations? This should be made clear.

**Other Strengths And Weaknesses:**

The paper presents a minimally invasive yet effective single-layer approach to unlearning that is novel in two different ways: they target only a single layer and also only perform a single step, which effectively linearizes the unlearning update. They specialize their method by instantiating it with CLIP-specific retain and forget losses, which still keeps it generally applicable to all CLIP-based models like Stable Diffusion or VLMs like LLaVa-1.5. This sets it apart from many related works in the area of unlearning in multimodal models, which often specialize their method on a particular model type. Moreover, the teaser figure is well-designed and effective in bringing across the main concepts of their method. Their perspective on finding room for unlearning in the “null space” of the retain loss provides a strong intuition before they introduce their method.

My main concerns with this work are the lack of depth in experimental evaluations (see Methods And Evaluation Criteria) as well as a lack of methodological ablations to analyze their approach and underline their design choices (e.g., what happens if only the importance but not the alignment measure is used to identify the single layer; what if SLUG would identify more than a single layer to strengthen its unlearning accuracy; what if the pareto-optimal pre-selection would be replaced by a random selection?).

Besides that, some aspects of the evaluations were unclear to me, most importantly the exact choice of FA and TA in the different scenarios and the compute requirements of performing those evaluations as part of their method. This appears crucial to me for a fair comparison to related methods. The existence or non-existence of a validation set for the inner evaluations is also something that should be clarified. A brief discussion around the limitations of SLUG, including the potential shortcomings as a single-layer approach, is also missing.

**Questions For Authors:**

n/a

**Relation To Broader Scientific Literature:**

Their proposed SLUG algorithm is a saliency-based machine unlearning method that is less restricted to a specific type of model (generality) than, e.g.,  ESD [3] or MACE [4] which are restricted to only diffusion models. SLUG is specifically instantiated for CLIP. which makes it more flexibly applicable to related models as the authors also try to transport in their paper by testing it on various CLIP-based models such as Stable Diffusion or a VLM. Their method is in its nature close to SalUn, which also tries to identify a portion of the parameters that are the most important for the update, but SalUn [1] only takes the “forget loss” into account to find the parameters that are the most important for the unlearning without any “alignment” considerations to incorporate the retain gradients into this selection. Instead of using a binary mask (like SalUn), SLUG goes further in a sense and identifies a single layer to restrict the unlearning to.

[1] Fan, Chongyu, et al. "Salun: Empowering machine unlearning via gradient-based weight saliency in both image classification and generation." arXiv preprint arXiv:2310.12508 (2023).

[2] Foster, Jack, Stefan Schoepf, and Alexandra Brintrup. "Fast machine unlearning without retraining through selective synaptic dampening." Proceedings of the AAAI conference on artificial intelligence. Vol. 38. No. 11. 2024.

[3] Gandikota, Rohit, et al. "Erasing concepts from diffusion models." Proceedings of the IEEE/CVF International Conference on Computer Vision. 2023.

[4] Lu, Shilin, et al. "Mace: Mass concept erasure in diffusion models." Proceedings of the IEEE/CVF Conference on Computer Vision and Pattern Recognition. 2024.

**Theoretical Claims:**

The sentence “Small alignment between unlearn and retain gradients would prevent unlearning updates from negatively affecting the retain set” in Sec. 2.2 needs more justification. Intuitively, if both of these gradients are exactly the same and thus perfectly aligned, there would not be any negative interference between the two, right? Another paragraph here or a visualization could help to transport the intuition of the authors to the reader.

---

> ### Author Rebuttal · Authors · 2025-04-01
>
> We appreciate the reviewer's thoughtful comments. We address the concerns as following:
>
> ## SOTA Results on UnlearnCanvas
> Thank you for your suggestion of reporting a unified summarizing metric (e.g., harmonic-like mean).
>
> To unify the scores of different metrics, we use the mean Gap Ratios (GP), defined as: $\frac{|\mathrm{current method performance} - \mathrm{best method performance}|}{\mathrm{best method performance}}$. For example, according to Table 2, the FID GP for SLUG is calculated as: $= \frac{75.97-54.21}{54.21}=0.4$.
>
>
> We compute the GP across effectiveness, FID, time, and memory+storage in Table 2 and report the arithmetic mean of these ratios across these metrics in the following table. In Table R3, SLUG demonstrates the lowest GP among all methods, further supporting our claim that SLUG achieves a well-balanced trade-off between effectiveness and efficiency.
>
> Table R3: Gap ratio (%) averages of different unlearning methods over UnlearnCanvas metrics. Low value means smaller performance gap between the best performing method.
> | Method | SLUG | ESD | FMN | UCE | CA | SalUn | SEOT | SPM | EDiff | SHS |
> |:---:|:---:|:---:|:---:|:---:|:---:|:---:|:---:|:---:|:---:|:---:|
> | Gap ratio mean (↓) | 13 | 4063 | 370 | 342 | 524 | 622 | 71 | 19042 | 1191 | 1013 |
>
> ## Adversarial Robustness
> Addressed in response to **Reviewer Ceck, Table R2**
>
> ## Methods and Evaluation Criteria
>
> ### Identity Erasure Details
> FA columns in Table 1 show the average forget accuracy for unlearning a single identity at a time (score averaged over 5 identities from Figure 3). TA columns report test accuracy on 100 celebrities from CelebA sampled as our validation set. For each identity unlearning experiment, we used forget sets of 1,000-6,000 image-text pairs associated with that identity. We will be happy to further clarify this in the text.
>
> ### Object Unlearning
> Our object unlearning is not restricted to specific ImageNet classes. Instead, we leverage the latest UnlearnCanvas, covering 20 object classes for a unified and comprehensive comparison. Our Table 4 focus on CLIP unlearning, differs from Table 2 in SalUn on SD unlearning.
>
> ### VLM Metrics
> We use forget accuracy (FA) to measure successful unlearning of target identities while using standard VLM benchmarks (MME, GQA, MMBench) to evaluate utility preservation. Table 3 shows we reduced FA to an average of 2.8% while maintaining high benchmark scores within 2-4% of original performance. This comprehensive evaluation demonstrates both effective unlearning and utility preservation.
>
> ## Theoretical Claims
> In this paper, we do not make any theoretical claims
>
> ## Evaluation Function and Costs
> The eval function measures both forget accuracy (FA) and test accuracy (TA) on **validation sets** distinct from the forget/retain training sets. A small set of validation images is a sufficient indicator of the unlearning step-size search process.
> - For CLIP, we used 5% of the test size
> - For SD, we used 10 test-time generated images not in the (forget) training set
> - For VLM, we used 10-image subset per identity for validation
>
> Our approach isn't "multi-step training" but rather a principled search for the optimal operating point along a single gradient direction, which is fundamentally different from iterative optimization methods.
>
>
> ### Computation Requirements
> The binary search evaluation typically requires less than 5 forward passes through the model (no backward passes) to identify the optimal step size. These costs are included in our reported computational efficiency metrics in Table 2 and are significantly lower than iterative methods requiring multiple forward-backward passes.
>
> ## Relation to Broader Scientific Literature
> We have acknowledged the connections to SalUn and SSD, and discussed them in Section 2 when introducing our approach.
>
> ## Methodological Ablations
>
> ### Importance-only, Random Selection
> Please refer to response to **Reviewer Ceck, Table R1**
>
> ### Multiple Layers vs Single Layer
> Following Table 1 setup, Table R4 presents updating multiple layers on the Pareto front and all model layers. While this slightly improved unlearning effectiveness (~2-3% better FA), it significantly increased complexity and utility degradation risk.
> Figure 8 shows that our approach effectively unlearns multiple concepts by targeting multiple layers in parallel, demonstrating the model’s ability to support modular unlearning without requiring multiple layers for a single concept.
>
> Table R4: Additional studies on updating multiple layers for CLIP unlearning.
> | Method | FA@1 (↓)  | FA@5 (↓)  | TA_IN@1 (↑)  | TA_CA@1 (↑) |
> |---|---|---|---|---|
> | All Pareto | 0.00  | 0.00 | 59.92 | 51.64 |
> | All Layers | 0.00  | 0.00 | 59.70 | 53.74 |
> | SLUG | 0.00 | 0.00  | 59.96  | 58.32 |

---

> > ### Comment · Reviewer_oTLd · 2025-04-03
> >
> > I like to thank the authors for their detailed rebuttal and effort in addressing my comments. I appreciate the authors’ additional explanation on the validation sets, which clarified the concerns I initially had about that. It’s much appreciated that the authors added Table R1 with a small ablation study in the CLIP unlearning scenario over 5 different celebrity targets. It’s a limited comparison with ablated versions, but still gives a good impression. The included comparison to the SalUn selection mechanism especially helps, as it shows that the final SLUG approach can outperform it. The additional small experiments on robustness against adversarial inversion attacks (P4D, UnlearnDiffAtk) reveal that SLUG also struggles with those, which is okay as long as they don’t claim that SLUG is robust against it (which they don’t).
> >
> > My questions on the compute requirements are now mostly clarified; they convinced me with the argument that only 5 forward (no backward) passes are typically sufficient in finding a good step size for the update, together with their provided details on the existence and scale of the validation sets.
> >
> > ### Remaining Questions and Concerns
> > 1. However, I would appreciate it if they could comment on how much the runtime / effectiveness of SLUG depends on the choice/size of the validation set. Did they build an intuition here on the trade-off? Did they try more thorough validations in-the-loop at the cost of efficiency but gaining a better estimate of the Pareto frontier and vice versa?
> >
> > 2. There is another minor concern from my initial review left: they mentioned that they added a Mean Gap Ratios (GP) Table R3 across effectiveness, FID, time, and memory+storage and then took the arithmetic mean of them. Even though I appreciate them making an effort to add a summarizing metric, I am questioning this choice since different metrics, especially the time metric, have different scales and dynamics. I see that they divide by the best metric value per column to have a normalizing effect, but still I think the message can benefit from leaving out the efficiency metrics and only compute the summarizing GP mean over the effectiveness metrics. SLUG is already clearly superior to the other approaches on that side of the metrics so there is no need to complicate the GP mean by including the efficiency metrics that can distort the picture a bit. Please also double-check that the values correctly computed in Table R3 as I quickly tried recreating them, which worked for the example that was provided in the text, but led to slightly different numbers for the ones reported in the table.
> >
> > 3. In addition to the above, I agree with the points made by Reviewer C1HE that a discussion on the advantages and disadvantages that come with the single-layer single-update approach is missing and should be addressed. Specifically, the lack of robustness and sometimes inferior unlearning accuracy compared to other approaches should be emphasized alongside the obvious benefits of a highly efficient, minimally invasive approach that the authors present.
> >
> > ### Conclusion:
> > Overall, the authors addressed my major concerns. In the expectation that the authors will revise the paper according to the rebuttals provided to all reviewers and include the additional results, I raise my recommendation from 2 to 3 (Weak Accept).
> > Their results with identifying and updating only a single layer for unlearning is a noteworthy contribution to the field that will be valuable to others.

---

> > > ### Author Response · Authors · 2025-04-05
> > >
> > > Thank you very much for acknowledging our efforts and increasing your score. We will certainly revise the paper to include all the results and points discussed during the rebuttal. We also appreciate your insightful questions and answer them below.
> > >
> > > ## Validation size, runtime, effectiveness
> > > A single `eval` function runtime increases **linearly with the validation set size**. In Table R6, we provide the **eval runtime** and **effectiveness of SLUG** versus different validation set sizes, following the setup of Table 1.
> > > Note that our original choice of 5% validation size already provides a good test accuracy on ImageNet, close to that of the original model (which achieves 60.12%). While increasing the validation size slightly improves utility retention after unlearning, it also increases evaluation time proportionally. Furthermore, a smaller validation size (1%) reduces the eval time to ~3 seconds at the expense of slightly reduced TA.
> > >
> > > Table R6: Forget accuracy, test accuracy on ImageNet, and runtime of unlearned CLIP models under various validation sizes.
> > > | Validation size | Number of images | Eval function cost (s) | FA@1 (↓)  | TA_IN@1 (↑)  |
> > > |:---:|:---:|:---:|:---:|:---:|
> > > | 1% | 500 | 3.05 | 0.0 | 58.83 |
> > > | 5 % (original) | 2500 | 6.62 | 0.0 | 59.96  |
> > > | 20 % | 10000 | 24.83 | 0.0 | 59.94 |
> > > | 50 % | 25000 | 59.69 | 0.0 | 59.98 |
> > > | 100 % | 50000 | 119.04 | 0.0 | 60.04 |
> > > ## Unified metric for UnlearnCanvas
> > > ### Mean Gap Ratios of Effectiveness metrics
> > > We appreciate your comments about the summarizing metrics. Your question prompted us to further analyze the summary metrics.
> > >
> > > Table R7 provides the summary statistics of the Gap Ratio (GP) using only the seven effectiveness metrics (i.e., {UA, IRA, CRA}_{style, object}, and FID). We compute GP for each metric independently (which is equivalent to centering and normalizing each metric for all methods, as you rightly pointed out) to get a 7-dimensional GP vector for each method. We can then summarize the GP vectors using some appropriate norm, which would measure the distance of every method from the “hypothetical reference model” that has the best performance of all metrics. We report the summary metrics using an L1 norm (proportional to mean GP) and an L2 norm (both divided by 7–the vector length).
> > >
> > > We note that while SLUG is not the best-performing method under effectiveness-only metrics, it offers competitive performance: 2nd best after SalUn in L2 norm by a small margin. We are not advocating for either metric here, and admit that different choices of norms and weighting of individual metrics can provide us different results for the summary metric. The two summary metrics offer a more comprehensive geometric characterization of the performance gap for each method. We hope this perspective will help the readers, and we thank you for raising these questions.
> > >
> > >
> > > Table R7: Gap ratio summary of different unlearning methods over effectiveness metrics. Low value means smaller performance gap from the best performing method. SLUG is competitive in effectiveness-only metrics as well.
> > > | Method | SLUG | ESD | FMN | UCE | CA | SalUn | SEOT | SPM | EDiff | SHS |
> > > |:---:|:---:|:---:|:---:|:---:|:---:|:---:|:---:|:---:|:---:|:---:|
> > > | Gap ratio L1 norm (↓) | 0.216 | 0.264 | 0.308 | 0.305 | 0.271 | 0.138 | 0.310 | 0.246 | 0.209 | 0.245 |
> > > | Gap ratio L2 norm (↓) | 0.100 | 0.137 | 0.134 | 0.155 | 0.138 | 0.098 | 0.158 | 0.121 | 0.114 | 0.111 |
> > >
> > > ### Table R3 clarification
> > > In Table R3, we computed the mean GP over “average accuracy”, FID, Time, and “average memory”. In other words, we first computed the average of 6 accuracy values (i.e., Average(UA, IRA, CRA)_{style, object}) and then computed the GP over the average accuracy for each method. We also computed the average of memory and storage columns and then computed GP for each method. Finally, we computed the mean of GP for average accuracy, FID, Time, and average memory (and reported as a percentage). We apologize for the confusion with Table-R3 and agree with you that leaving the time and memory out of the summary metrics offers a clear comparison for effectiveness.
> > >
> > > ## Discussion on advantages and disadvantages of the single-layer approach
> > > We will certainly add the discussion on the advantages and disadvantages of our approach along with the ablation studies in the revision.
> > >
> > > We hope our comment addresses some of your concerns and will be happy to further discuss if something remains unclear.

---

### Official Review · Reviewer_Ceck · 2025-03-18

**Overall Recommendation:** 4

**Summary:**

This paper introduces SLUG, an efficient targeted unlearning method that aims to remove specific unwanted information from large-scale models with minimal computational overhead. Unlike conventional unlearning approaches that iteratively update parameters across the entire model, SLUG identifies a single critical layer using metrics of layer importance and gradient alignment. By performing only one-time gradient computations on the forget and retain losses, and updating that selected layer along a linear trajectory (with a carefully chosen step size via binary search), SLUG achieves effective unlearning while preserving the model’s utility. The method is evaluated on a diverse set of models including CLIP, Stable Diffusion, and vision-language models, and experiments on the UnlearnCanvas benchmark demonstrate that SLUG delivers comparable unlearning performance to existing methods but with significantly lower computational cost.

**Claims And Evidence:**

Yes.

**Essential References Not Discussed:**

Please refer to weakness part.

**Experimental Designs Or Analyses:**

Yes

**Methods And Evaluation Criteria:**

Yes, it mainly follows previous benchmark settings.

**Other Comments Or Suggestions:**

Please refer weakness part.

**Other Strengths And Weaknesses:**

Strengths:
- SLUG’s one-time gradient computation and single-layer update significantly reduce the computational burden compared to iterative unlearning methods. This efficiency is particularly valuable when dealing with large-scale models.

- The idea of focusing on a single layer, identified via layer importance and gradient alignment metrics, for unlearning is both innovative and practical. Concentrating updates in a minimal subset of the model helps limit unintended side effects on overall model performance.

Weakness:

- Limited Domain Generalization: Although SLUG is evaluated on vision-language and diffusion models, it remains unclear how well the single-layer update approach generalizes to other domains, such as text-only large language models. Additionally, since previous work has used established benchmarks (e.g., for NSFW removal tasks on diffusion models), it would be beneficial to include performance comparisons on those benchmarks to better contextualize the results.

- Lack of Ablation Studies: The paper does not include ablation studies comparing different weight identification methods in combination with the single gradient direction unlearning approach. For instance, if an alternative method like SalUn were applied with a single gradient update, resulting in a comparable time complexity of $O(2·N_f+ N_r)$, it would be informative to assess whether a finer-grained weight localization can yield improved performance.

- Limited Robustness Evaluation: While the authors present one robustness evaluation on diffusion models, incorporating additional evaluation methods (e.g., CCE [1] and UnlearnDiffAtk [2]) would provide a more comprehensive assessment of the method’s resilience under various adversarial or challenging conditions.

[1] Pham M, Marshall K O, Cohen N, et al. Circumventing concept erasure methods for text-to-image generative models[J]. arXiv preprint arXiv:2308.01508, 2023.

[2] Zhang Y, Jia J, Chen X, et al. To generate or not? safety-driven unlearned diffusion models are still easy to generate unsafe images... for now[C]//European Conference on Computer Vision. Cham: Springer Nature Switzerland, 2024: 385-403.

**Questions For Authors:**

Please refer weakness part.

**Relation To Broader Scientific Literature:**

Related to the trustworthy machine learning

**Theoretical Claims:**

No theoretical claims.

---

> ### Author Rebuttal · Authors · 2025-04-01
>
> Thank you for summarizing the strengths of SLUG (computational efficiency and innovative single-layer update approach). We address your concerns below:
>
> ## Domain Generalization
> While our paper primarily focuses on CLIP, Stable Diffusion, and VLMs, the principles behind SLUG are agnostic to domains and applicable to LLMs. Nevertheless, a detailed analysis of SLUG for LLMs would require a non-trivial time and effort (due to differences in task and dataset). We believe our comprehensive evaluation across multiple multi-modal foundation models already demonstrates the versatility and effectiveness of our method across different domains
>
>
> ## Evaluation on NSFW Benchmark
> The adversarial attacks evaluation in [2] employed NSFW and object unlearning scenarios. We applied SLUG to unlearn SDv1.4 on the "Nudity" concept and "Church" object, using 142 and 50 author-provided prompt-generated images, respectively, as forget sets.
> We present the results in Table R0, where we have the lowest (best) forget accuracy, which confirms that SLUG is generalizable to unlearning concepts and objects previously studied in prior work. We will be happy to expand this evaluation for the complete dataset in the revision.
>
> Table R0 - Additional evaluation on unlearning Nudity (NSFW) and Church (Object) scenarios.
> | Concept/Object |  | Nudity (↓)|  |  | Church (↓)|  |
> |:---:|:---:|:---:|:---:|:---:|:---:|:---:|
> | Unlearn Method | SLUG | ESD | FMN | SLUG | ESD | FMN |
> | Forget Accuracy (%) | 16.90 | 20.42 | 88.03 | 4 | 14 | 52 |
>
> ## Ablation Studies
> In Table R1, we adopt the weight selection scheme in SalUn (i.e., selecting a mask across the entire network). Additionally, we conduct an ablation study on our introduced metrics: layer importance and gradient alignment, and randomly selected layer, to provide further insights.
>
> Table R1 follows the setup in Table 1, focusing on unlearning CLIP and evaluating forget accuracy on 5 target identities, and zero-shot test accuracy on ImageNet and CelebA.
> - "SalUn": Selecting model weights distributed across the network (instead of a single layer) using importance-only (SalUn-like selection) performs worse than SLUG, with higher  FA and lower TA.
> - "Importance": Selecting a single layer using only gradient importance results in FA reduction to 0 but TA_IN and TA_CA also reduced, due to some layers with high importance for unlearning exhibit high gradient conflict with the retain set.
> - "Alignment": Selecting a single layer based on gradient alignment only results in increased FA@5 of 5.56% as well as reduction in TA.
> - ”Random”: Selecting a single layer at random performs poorly, with an increase in FA and a decrease in TA compared to SLUG.
>
> These results further demonstrate that both importance and alignment are required for achieving an optimal trade-off between effective unlearning and utility retention.
>
> Table R1: Additional studies on different parameter selection approaches for CLIP unlearning.
> | Parameter selection | FA@1 (↓)  | FA@5 (↓)  | TA_IN@1 (↑)  | TA_CA@1 (↑) |
> |:---:|:---:|:---:|:---:|:---:|
> | “SalUn” (distributed weights, importance only)  | 4.44 | 11.33 | 48.23 | 37.38 |
> | Single layer importance only | 0.0 | 0.0 | 21.04 | 42.00 |
> | Single layer alignment only | 0.0 | 5.56 | 31.08 | 54.16 |
> | Single layer at random | 0.0 | 6.91 | 33.38 | 52.90 |
> | SLUG | 0.00 | 0.00  | 59.96  | 58.32 |
>
> ## Adversarial Robustness Evaluation
> Our discussions and experiments in Sec. D.1 was limited to robustness against (prompt-based) concept arithmetic attacks and quantization. Guaranteeing robustness against whitebox attacks is challenging, and we did not make any such claim in our paper. Nevertheless, we appreciate the suggestion to incorporate adversarial evaluations. In Table R2, we utilize the latest UnlearnDiffAtk [2] and P4D [1]. Specifically, we selected the "Nudity" and "Church" categories from Table 2 and Table 4 in [2] to provide a brief adversarial evaluation of SLUG.
>
> Following the same setup as [2], we applied SLUG on SDv1.4 to unlearn "Nudity" concept and "Church" object, then attacked the SLUG-unlearned SDv1.4 using two attack methods: UnlearnDiffAtk and P4D that optimized 142 and 50 adversarial prompts for "Nudity" and "Church," respectively.
>
> The results indicate that SLUG (like other unlearning methods) **is not immune to whitebox adversarial attacks**.
>
> Table R2: Evaluation against adversarial attacks. Lower ASR (%) indicates better adversarial robustness. Row-“No Attack” shows the original performance on unlearning tasks.
> | Concept/Object |  | Nudity (↓)|  |  | Church (↓)|  |
> |:---:|:---:|:---:|:---:|:---:|:---:|:---:|
> | Attack method | SLUG | ESD | FMN | SLUG | ESD | FMN |
> | No Attack | 16.90 | 20.42 | 88.03 | 4 | 14 | 52 |
> | P4D | 76.76 | 69.71 | 97.89 | 46 | 56 | 98 |
> | UnlearnDiffAtk | 90.32 | 76.05 | 97.89 | 80 | 60 | 96 |
>
>
> [1] Prompting4Debugging (ICML 24’)
>
> [2] To generate or not? (ECCV 24’)

---

> > ### Comment · Reviewer_Ceck · 2025-04-05
> >
> > Thank you for providing a detailed rebuttal. The additional experiments addressed most of my concerns, and I have therefore raised my score to 4.

---

> > > ### Author Response · Authors · 2025-04-05
> > >
> > > Thank you very much for acknowledging our efforts and increasing the score!

---

### Decision · Program_Chairs · 2025-05-01

**Decision:**

Accept (poster)

**Comment:**

This paper introduces an unlearning algorithm, SLUG, that promises removal of specific information with significantly lower computational overhead compared to some other methods, while preserving model utility. SLUG achieves this by efficiently identifying a single layer to update. Reviewers initially raised several concerns, such as robustness of unlearning to white box adversarial attacks, and the scope of experiments, which were mostly addressed during the rebuttal process with new experiments. In the camera-ready version, the authors should incorporate these new results, improve their discussion of computational costs, trade-offs, and single-layer editing approach, and also accurately state the limitations of their robustness evaluations (and consider expanding these evaluations).